# Metformin reverses early cortical network dysfunction and behavior changes in Huntington's disease

Isabelle Arnoux[1†], Michael Willam[2†], Nadine Griesche[3†], Jennifer Krummeich[2], Hirofumi Watari[1], Nina Offermann[3], Stephanie Weber[3], Partha Narayan Dey[4], Changwei Chen[5], Olivia Monteiro[5], Sven Buettner[3], Katharina Meyer[3], Daniele Bano[3], Konstantin Radyushkin[6], Rosamund Langston[5,6], Jeremy J Lambert[5], Erich Wanker[7], Axel Methner[4‡], Sybille Krauss[3‡*], Susann Schweiger[2‡*], Albrecht Stroh[1‡*]

[1]Institute of Pathophysiology, Focus Program Translational Neurosciences, University Medical Center, Mainz, Germany; [2]Institute for Human Genetics, University Medical Center, Mainz, Germany; [3]German Center for Neurodegenerative Diseases (DZNE), Bonn, Germany; [4]Department for Neurology, University Medical Center, Mainz, Germany; [5]Division of Neurosciences, Ninewells Hospital and Medical School, Dundee, United Kingdom; [6]Mouse Behavior Unit, University Medical Center, Mainz, Germany; [7]Department of Neuroproteomics, Max-Delbrück-Center, Berlin, Germany

**\*For correspondence:**
Sybille.krauss@dzne.de (SK);
schweigs@uni-mainz.de (SS);
albrecht.stroh@unimedizin-mainz.de (AS)

[†]These authors contributed equally to this work
[‡]These authors also contributed equally to this work

**Competing interests:** The authors declare that no competing interests exist.

**Abstract** Catching primal functional changes in early, 'very far from disease onset' (VFDO) stages of Huntington's disease is likely to be the key to a successful therapy. Focusing on VFDO stages, we assessed neuronal microcircuits in premanifest Hdh150 knock-in mice. Employing *in vivo* two-photon $Ca^{2+}$ imaging, we revealed an early pattern of circuit dysregulation in the visual cortex - one of the first regions affected in premanifest Huntington's disease - characterized by an increase in activity, an enhanced synchronicity and hyperactive neurons. These findings are accompanied by aberrations in animal behavior. We furthermore show that the antidiabetic drug metformin diminishes aberrant Huntingtin protein load and fully restores both early network activity patterns and behavioral aberrations. This network-centered approach reveals a critical window of vulnerability far before clinical manifestation and establishes metformin as a promising candidate for a chronic therapy starting early in premanifest Huntington's disease pathogenesis long before the onset of clinical symptoms.
DOI: https://doi.org/10.7554/eLife.38744.001

## Introduction

Huntington's disease is caused by the expansion of a CAG repeat in the open-reading frame of the huntingtin gene (*HTT*), which translates into an expanded glutamine stretch in the aberrant, mutant protein (mHTT). Huntington's disease has primarily been described as a late-onset neurodegenerative disease. However, it is preceded in its premanifest period by a prolonged presymptomatic phase followed by a prodromal phase with hardly detectable and unspecific symptoms occurring far before classical Huntington's disease becomes apparent (*Ross et al., 2014*). These symptoms include reduced impulse control, social disengagement, low conversational participation, reduction of the concentration span and decline of clearly defined cognitive domains (*Stout et al., 2011*; *Predict-HD Investigators of the Huntington Study Group et al., 2007*; *Ross et al., 2014*; *Labuschagne et al., 2016*) and are accompanied by slight changes of cortical

**eLife digest** Huntington's disease is a devastating brain disorder that causes severe mood disorders, problems with moving, and dementia. Most people develop the condition between their thirties and fifties, and die a decade or two after the symptoms first appear.

The disease emerges because of a mutation in the gene for the Huntingtin protein, which leads to neurons slowly dying in the brain. While genetic testing can reveal who carries the faulty gene, no treatment addresses the root of the disorder or prevents it from appearing. Instead, most therapies for Huntington's disease aim to reduce brain damage once the telltale symptoms are already present. However, the disease-causing protein is expressed early during the life of a patient, which could give it time to damage the brain long before neurons die and the disorder reveals itself. Treatments that start after the first signs of the disease may be too late to reverse the damage. Detecting and preventing early brain changes in people that carry the mutation may thus help to stop the disease from progressing.

Here, Arnoux, Willam, Griesche et al. set out to detect the minute changes that the faulty Huntingtin protein may cause in the brain network of young mice with the mutation. State-of-the-art imaging tools helped to examine individual neurons in the brain area that processes visual information. These experiments revealed that a group of brain cells had become hyperactive; once this change had occurred, the mutant animals were less anxious than is typical for mice.

Metformin is a drug used to treat diabetes, but it also interferes with a structure that is required to produce the disease-causing Huntingtin protein. Arnoux et al. therefore explored whether the compound could rescue the early brain alterations observed in mutant mice. Adding metformin in the water of the animals for three weeks halted the production of the mutant protein, reversed the brain changes and stopped the abnormal behavior.

Further work is now required in humans to confirm that Huntington's disease starts with a change in the activity of networks in the brain, and to verify that metformin can stop the disorder in its track.
DOI: https://doi.org/10.7554/eLife.38744.002

network topology and functional connectivity in resting state fMRI measures (*PREDICT-HD investigators of the Huntington Study Group et al., 2015*; *Wolf et al., 2012*). Importantly, such subtle network dysregulations may occur even earlier than the described prodromal symptoms in a very far from disease onset (VFDO) premanifest stage (*Figure 1a*). This stage reaches back more than 15–20 years before motor symptoms become visible in patients and far before protein aggregates and neurodegeneration are observed.

In Huntington's disease occurrence of such very early changes is supported by the observation of early deficits in premanifest Huntington's disease mutation carriers, such as loss of phosphodiesterase 10A in the occipital lobe up to 47 years prior to disease onset (summarized in *Wilson et al., 2017*). Also, the ability to perform complex visuospatial orientation, such as visual search, seems to be altered even in pre-manifest stages far from clinical diagnosis (*Labuschagne et al., 2016*). We hypothesize that primal functional changes in the VFDO stage of premanifest Huntington's disease open up very early vulnerable windows for disease development and preventive therapy prior to neuronal loss and may also provide promising early biomarkers for therapy development (*Mehler and Gokhan, 2000*; *Kerschbamer and Biagioli, 2015*; *Humbert, 2010*; *Cepeda et al., 2003*).

It seems that the visual cortex is one of the first regions that are functionally affected during disease development in Huntington's disease (*Labuschagne et al., 2016*; *Dogan et al., 2013*). In order to identify primal network changes, we have here established a network-centered approach and focused on microcircuit function in layer 2/3 of the visual cortex at an early premanifest stage in a mouse model of Huntington's disease corresponding to the VFDO stage in premanifest Huntington's disease. We have used two-photon imaging using fluorescent indicators of intracellular $Ca^{2+}$ in order to resolve the functional architecture of intact cortical microcircuits *in vivo* with single neuron resolution (*Grienberger and Konnerth, 2012*). We show that even at the early age of 10 – 15 weeks (young adults compared to humans), the entire cortical microcircuit shifts towards a more excitable

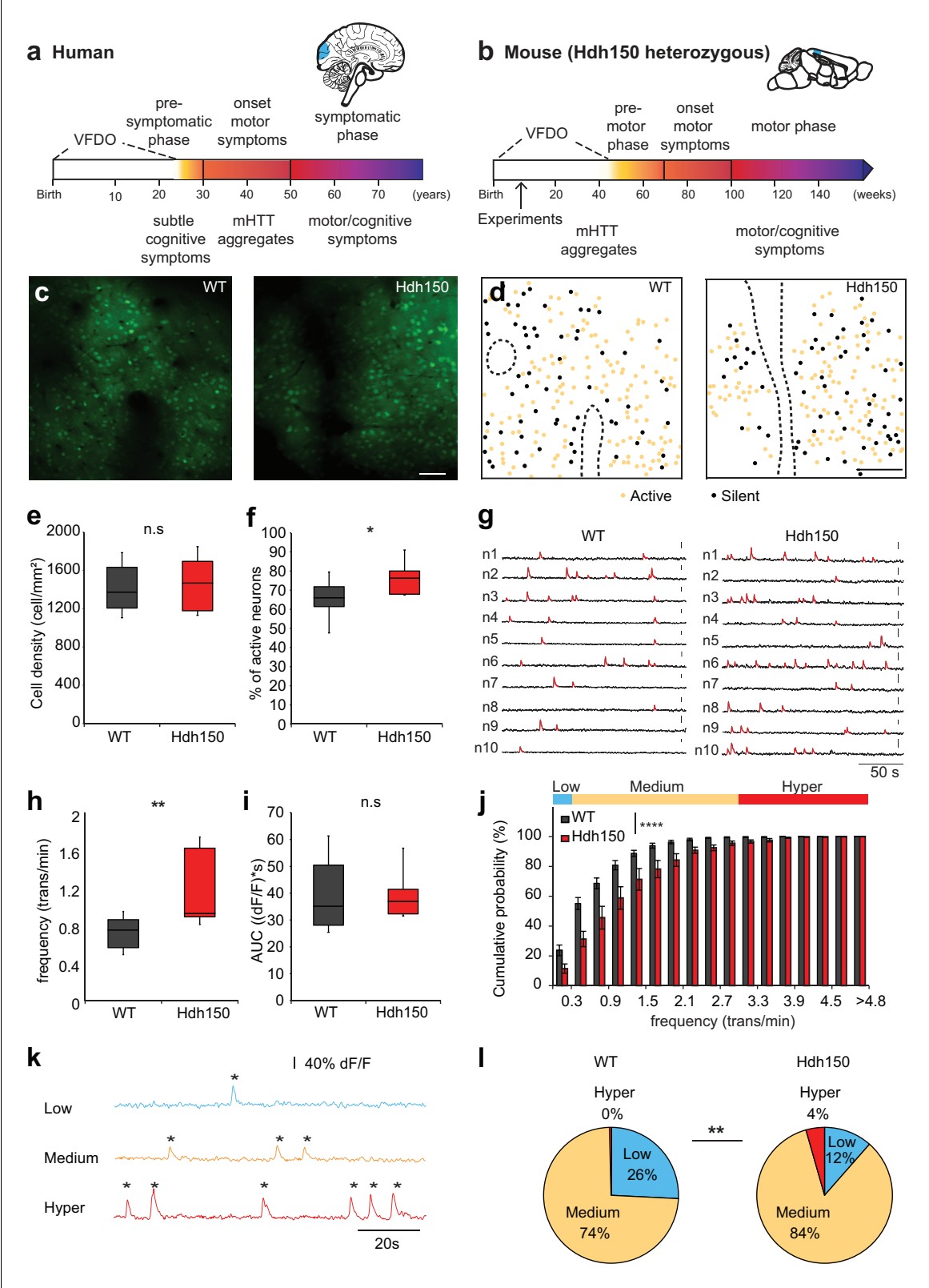

**Figure 1.** *In vivo* two-photon Ca$^{2+}$ imaging in layer 2/3 of visual cortex reveals a hyperactive neuronal activity pattern prior to disease onset. (a,b) Top right, Illustrations indicating the visual cortex (blue area) in human (a) and mouse (b) brains. The brains are not drawn to scale. Bottom, timeline of Huntington's disease progression in human and Hdh150 mouse model of Huntington's disease. The Huntington's disease onset was age 30–50 years in humans and ~70 weeks in Hdh150 mice. We conducted our experiments during a very early pre-symptomatic phase, far prior to mHtt aggregates and

*Figure 1 continued on next page*

*Figure 1 continued*

motor symptoms. VFDO: very far from disease onset. (**c**) Representative two-photon images of OGB-1 AM staining in layer 2/3 of the visual cortex of WT and Hdh150 mice. Scale bar: 70 μm. (**d**) Color-coded maps of silent (black) and spontaneously active (orange) neurons in WT (left) and Hdh150 (right) mice. Dashed lines represent the boundaries of blood vessels (original images in *Figure 1c*). Scale bar: 100 μm. (**e**) Density of stained cells in layer 2/3 of the visual cortex in WT and Hdh150 mice. Unpaired t-test, p=0.71. (**f**) Increased proportion of spontaneously active neurons in Hdh150 mice. Unpaired t-test, p<0.05. (**g**) Representative traces of spontaneous $Ca^{2+}$ transients (red) of 10 neurons recorded in vivo in WT and Hdh150 mice. Vertical scale bars: 40% dF/F. (**h**) Increased frequency of $Ca^{2+}$ transients in spontaneously active neurons of Hdh150 mice. Silent neurons were excluded, as in subsequent panels. Mann-Whitney test, p<0.01 (**i**) Quantification of area under the curve (AUC) of $Ca^{2+}$ transients. Unpaired t-test, p=0.98 (**j**) Cumulative frequency distribution of $Ca^{2+}$ transients in WT (dark grey) and Hdh150 (red) mice. Top, colored categorization of neurons according to their $Ca^{2+}$ transient frequencies. Two-way ANOVA, group: p<0.0001, time: p<0.0001, Interaction: p<0.0001. (**k**) Color-coded categorization of neurons according to their $Ca^{2+}$ transient frequency: 'low' (<0.3 trans/min, blue; silent neurons excluded), 'medium' (0.3–3 trans/min, orange) and 'hyper' (>3 trans/min, red). Each peak is marked by an asterisk. (**l**) Relative proportion of low, medium and hyperactive neurons in layer 2/3 of the visual cortex in WT (left) and Hdh150 (right) mice. Chi-square test, p<0.01.

DOI: https://doi.org/10.7554/eLife.38744.003

The following video, source data, and figure supplements are available for figure 1:

**Source data 1.** Numerical values of *Figure 1* and associated supplement figures.
DOI: https://doi.org/10.7554/eLife.38744.007
**Source data 2.** Code used for the analysis of calcium imaging.
DOI: https://doi.org/10.7554/eLife.38744.008
**Figure supplement 1.** $Ca^{2+}$ events from astrocytes and neurons show clearly distinct kinetics.
DOI: https://doi.org/10.7554/eLife.38744.004
**Figure supplement 2.** Cortical hyperactivity is independent of mHtt aggregation, astrogliosis or apoptotic cell death in presymptomatic VFDO Hdh150 mice.
DOI: https://doi.org/10.7554/eLife.38744.005
**Figure supplement 3.** Randomization of experimental data to assess specific spatial clustering.
DOI: https://doi.org/10.7554/eLife.38744.006
**Figure 1—video 1.** *In vivo* two-photon images of mouse visual cortex performed at different depths (indicated in the upper left corner) after multi-bolus loading with OGB-1 AM.
DOI: https://doi.org/10.7554/eLife.38744.009
**Figure 1—video 2.** Representative time-lapse of *in vivo* two-photon $Ca^{2+}$ imaging acquired in layer 2/3 of mouse visual cortex showing single-cell $Ca^{2+}$ transients.
DOI: https://doi.org/10.7554/eLife.38744.010

state characterized by a complex change in neuronal activity pattern and hyperactive neurons. These findings are accompanied by changes in animal behavior, including a decrease in anxiety.

No effective and curative treatment has been developed for Huntington's disease so far (*Frank, 2014*). A chronic drug therapy that commences early on in VFDO stages in premanifest Huntington's disease and ameliorates early dysregulations as the potential origin of pathogenic processes and disease spreading is therefore a promising and necessary strategy. In Huntington's disease animal models, even short-term reduction of protein load through RNA interference and antisense strategies has beneficial effects on disease phenotypes and progression lasting several months after intervention (*Stanek et al., 2014*; *DiFiglia et al., 2007*; *Keiser et al., 2016*). We have recently shown that mRNAs carrying CAG repeats bind to a protein complex containing the ubiquitin ligase midline 1 (MID1) in a repeat size-dependent manner. Through ubiquitination MID1 regulates PP2A (protein phosphatase 2A) and mTOR (mechanistic target of rapamycin) activities and the translation of associated mRNAs (*Krauss et al., 2013*; *Griesche et al., 2016*). Disruption of the MID1/PP2A/mTOR protein complex leads to an increase of PP2A activity, a decrease of mTOR activity and a reduction of translation rates of mRNAs with expanded CAG repeats (*Krauss et al., 2013*).

We show here that the type II diabetes drug metformin interferes with the MID1/PP2A/mTOR protein complex and significantly reduces the translation rate of *Htt* mRNA, resulting in a reduction of aberrant Htt protein production *in vitro* and *in vivo* in the Hdh150 mouse model. Notably, in Hdh150 mice *in vivo* metformin, when given early in the VFDO stage, and chronically in the drinking water, fully reverses both early neuronal network dysregulations and behavioral aberrations.

## Results

### Increase in cortical neuronal network activity in presymptomatic Huntington's disease mice

Since the visual cortex is one of the first regions affected by the disease (*Dogan et al., 2013*; *Labuschagne et al., 2016*), we focused on layer 2/3 of the visual cortex of ~12 weeks old, heterozygous knock-in mice expressing expanded Htt with 150 glutamine repeats (Hdh150), in the lightly anesthetized mouse. This time corresponds to the VFDO in presymptomatic Huntington's disease (*Figure 1a,b*). We focused our analysis on male mice, thereby minimizing the influence of hormonal fluctuations on network activity. This approach is in line with a recent study in the field of Alzheimer's disease (*Iaccarino et al., 2016*), using males only. We employed two-photon $Ca^{2+}$ imaging *in vivo* using the synthetic $Ca^{2+}$ indicator Oregon-Green BAPTA1 (OGB-1) AM to monitor the suprathreshold activity of a neuronal microcircuit with cellular resolution, typically comprising around 200 neurons (*Kerr et al., 2005*) (*Figure 1c*, *Figure 1—figure supplement 1a,b*). Events from astrocytes, which are also stained by OGB-1, were excluded from this neuronal network based on their temporal dynamics (*Figure 1—figure supplement 1c–f*, *Figure 1—source data 1*).

By imaging at different depths, we found a similar spatial extent of OGB-1 staining in the visual cortex of Hdh150 and control wild-type (WT) mice (*Figure 1—figure supplement 1b*). *Figure 1—video 1* shows two-photon images acquired from the pial surface illustrating a homogenous OGB-1 staining in layers 1, 2 and 3 of mouse visual cortex. We observed no difference in the density of stained cells (*Figure 1c and e*, *Figure 1—source data 1*, WT: $1413 \pm 74$ cells/mm$^2$ (n = 11 mice), Hdh150 $1456 \pm 90$ cells/mm$^2$ (n = 10 mice), Mann-Whitney test p=0.8).

To identify changes in microcircuit activity, we assessed spontaneous activity *in vivo* in the cortical microcircuit of WT and Hdh150 mice, which reliably reflects the functional microarchitecture of sensory cortical areas (*Miller et al., 2014*). *Figure 1—video 2* shows an example of *in vivo* two-photon $Ca^{2+}$ imaging exhibiting ongoing spontaneous activity in layer 2/3 of mouse visual cortex with single-cell resolution. Both WT and Hdh150 mice exhibited silent and active cells (*Figure 1d*), but a significantly higher proportion of active neurons was detected in Hdh150 mice, indicating a more active network (*Figure 1f*, *Figure 1—source data 1*, WT: $65.1 \pm 3.5\%$ (n = 8 mice), Hdh150: $77.1 \pm 3.5\%$ (n = 6 mice), unpaired t-test p<0.05). No spatial clustering of silent or active cells could be observed in either group (*Figure 1d*).

Next, we analyzed the frequency of $Ca^{2+}$ transients and activity patterns in the population of active neurons (*Figure 1—source data 2*). Notably, the frequency of $Ca^{2+}$ transients was significantly higher in Hdh150 compared to WT mice (*Figure 1g–h*, *Figure 1—source data 1*, WT: $0.74 \pm 0.06$ trans/min (n = 8 mice), Hdh150: $1.2 \pm 0.14$ trans/min (n = 6 mice), Mann-Whitney test, p<0.01). No difference was found in the mean area under the curve (AUC) of $Ca^{2+}$ transients between WT and Hdh150 mice (*Figure 1i*, *Figure 1—source data 1*, WT: $39.8 \pm 3.9$ (dF/f)*s, Hdh150: $39.9 \pm 3.6$ (dF/f)*s, unpaired t-test, p=0.98) indicating that on average, the mean number of underlying action potentials for each individual calcium transient was not different in Hdh150 mice.

We furthermore analyzed the distribution of neurons according to their $Ca^{2+}$ transient frequency. The cumulative probability distribution of the activity of individual neurons in Huntington's disease mice was shifted toward higher frequencies indicating that the overall neuronal activity transitioned toward a more excitable network (*Figure 1j*, *Figure 1—source data 1*, two-way ANOVA p<0.0001). We sub-classified the active neurons into three functional subgroups according to their transient frequency: low, medium and hyperactive (*Figure 1k*, *Figure 1—source data 1*). Notably, we identified a unique subgroup in VFDO Hdh150 mice: hyperactive neurons (*Figure 1l*, *Figure 1—source data 1*, Chi-square test p<0.01). This subgroup was accompanied by a reduction in the number of neurons with low activity, which corroborates the shift of the microcircuit activity.

In later stages of Alzheimer's disease, hyperactive cells were shown to cluster near amyloid plaques (*Busche et al., 2008*). A similar scenario might occur in Huntington's disease. To clarify whether the cortex of VFDO Hdh150 mice is affected by mHtt aggregates - a later stage Huntington's disease hallmark - we performed immunohistochemistry in Hdh150 and WT animals. We observed only diffuse and non-aggregated Htt immunoreactivity in the cytoplasm of neurons in cortical areas of both WT and Hdh150 mice (*Figure 1—figure supplement 2a*) in accordance with previous studies (*Lin et al., 2001*). In addition, no reactive astrocytes were observed (*Figure 1—figure supplement 2b*) and only few cells that were stained for activated caspase-3, an apoptotic marker, in cortical

areas of WT and Hdh150 mice (*Figure 1—figure supplement 2c*) as previously observed (*Yu et al., 2003*). In order to test whether hyperactive cells cluster in the Hdh150 mouse model, spatial distance between every pair of neurons was quantified (*Figure 1—figure supplement 3c,d*, *Figure 1—source data 1*). We observed no significant differences between the permutations of functional subgroups in premanifest Hdh150 and control mice, reflecting a rather homogenous distribution of all functional subgroups (*Figure 1—figure supplement 3c,d*, *Figure 1—source data 1*, Mann-Whitney, not significant, see *Table 1* for p-values). This finding was confirmed in randomized data in which cell location was kept but functional identity was randomly permutated (*Figure 1—figure supplement 3a,b and d*, *Figure 1—source data 1*). Taken together, clustering of hyperactive cells could not be observed in VFDO Hdh150 animals and cortical hyperactivity is independent of mHtt aggregation, apoptotic cell death or astrogliosis at this presymptomatic VFDO stage.

## Increased synchronicity and cortical dysfunction in premanifest VFDO Huntington's disease mice

An important aspect of neuronal information processing is the optimization of encoding strategies. In the visual cortex, encoding of information is characterized by sparse and precisely timed neuronal activity. Cortical activity is defined by transiently co-active ensembles of neurons acting as a functional unit (*Miller et al., 2014*). To capture these spatiotemporal dynamics in the microcircuit, we analyzed synchronicity of the transients between all pairs of neurons (*Figure 2a*) by calculating Pearson's correlation coefficient (Pearson's r) for every pair. First, we confirmed that the level of synchronicity within a healthy cortical microcircuit is drastically higher than the random synchronicity in shuffled data (*Figure 2b*, *Figure 2—source data 1*, WT: $0.024 \pm 0.005$, WT rand: $-0.0003 \pm 0.0003$, Mann-Whitney test $p<0.0001$). Notably, the synchronicity was even higher in Hdh150 mice compared to healthy controls (*Figure 2b*, *Figure 2—source data 1*, WT: $0.02 \pm 0.005$, Hdh150: $0.04 \pm 0.006$, Mann-Whitney test $p<0.05$).

Next, we compared the synchronicity level in all functional subgroups (*Figure 2c*, *Figure 2—source data 1*). The pairs involving the low activity subgroups (LL and LM) showed no differences in WT and Hdh150 mice (Mann-Whitney test, in LL $p=0.5$ and in LM $p=0.7$). However, Pearson's r increased significantly for the medium-to-medium (MM) pairs in Hdh150. Pearson's r was even higher for medium-hyperactive (MH) and hyperactive-hyperactive (HH) compared to low-low (LL) pairs (Mann-Whitney test compared to LL in Hdh150 mice: in MM $p<0.05$, in MH and HH $p<0.01$).

To assess whether the increased synchronicity in the VFDO Hdh150 mice occurred merely due to the higher number of transients, we used randomized data with unchanged frequency but temporally shuffled transients (*Figure 1—figure supplement 3a and b*). Pearson's r in the randomized data was nearly zero for all, including the high frequency subgroups in WT and Hdh150 mice (*Figure 2—figure supplement 1a and b*, *Figure 2—source data 1*). This finding argued against the possibility that the increased Pearson's r in the experimental data occurred by chance.

We next asked whether functional ensembles with a high level of synchronicity are located in spatial vicinity to each other, by testing whether Pearson's r changed with physical distance between the pairs of neurons (*Figure 2d*). An inverse linear relationship was observed between Pearson's r and the pairwise distance in both the WT and Hdh150 mice (*Figure 2d*, *Figure 2—source data 1*, two-way ANOVA $p<0.0001$, WT vs Hdh150) suggesting that two closely located neurons have a higher probability to fire together. This is consistent with similar findings in the forelimb motor cortex of head-restrained mice (*Dombeck et al., 2009*). Randomization of the data abolished the inverse relationship between Pearson's r and distance (*Figure 2e*, *Figure 2—source data 1*, two-way ANOVA $p=0.3$, WT rand vs Hdh150 rand).

Since many studies, especially in premanifest mutation carriers, have linked Huntington's disease pathology to changes in metabolism (*Damiano et al., 2010*; *Duan et al., 2014*; *Jin and Johnson, 2010*; *Labbadia and Morimoto, 2013*; *Mochel and Haller, 2011*), hyperactivity might mirror metabolic dysregulation in subgroups of cortical cells. We used mitochondrial respiration as a marker of metabolic functionality and quantified mitochondrial respiration using high-resolution respirometry of cortical tissues. Mitochondrial respiration was unchanged in cortical tissue of Hdh150 mice suggesting that the observed neuronal hyperactivity occurs prior to metabolic changes (*Figure 2—figure supplement 2a–c*, *Figure 2—source data 1*, Mann-Whitney test, not significant, see *Table 1* for p-values).

**Table 1.** Statistics

| Figure | Test | Values | N |
|---|---|---|---|
| *Figure 1e* | Unpaired t test, two-tailed | NS, p=0.71 | WT n = 11 mice; Hdh150 n = 10 mice |
| *Figure 1f* | Unpaired t test, two-tailed | p=0.023 | WT n = 1204 cells in eight mice; Hdh150 n = 933 cells in six mice |
| *Figure 1h* | Mann-Whitney test | p=0.006 | WT n = 765 cells in eight mice; Hdh150 n = 695 cells in six mice |
| *Figure 1i* | Unpaired t test, two-tailed | NS, p=0.98 | WT n = 765 cells in eight mice; Hdh150 n = 695 cells in six mice |
| *Figure 1j* | Two-way ANOVA test | Group: p<0.0001, Df = 1, F = 85.96, time: p<0.0001, Df = 16, F = 147, Interaction: p<0.0001, F = 4.9, Df = 16 | WT n = 765 cells in eight mice; Hdh150 n = 695 cells in six mice |
| *Figure 1l* | Chi-square test | p=0.002, df = 1, Chi-square = 9.127 | WT n = 765 cells in eight mice; Hdh150 n = 695 cells in six mice |
| *Figure 2b* | Mann-Whitney test | WT vs Hdh150 p=0.03; WT vs WT rand p<0.0001; Hdh150 vs Hdh150 rand p<0.0001 | WT n = 26126 Pearson's r in eight mice; Hdh150 n = 58050 Pearson's r in six mice |
| *Figure 2c* | Mann-Whitney test | WT vs Hdh150 mice: MM p=0.041 in Hdh150 mice; compared to LL: MM p=0.0496, MH p=0.005, HH p=0.009 | WT n = 26126 Pearson's r in eight mice; Hdh150 n = 58050 Pearson's r in six mice |
| *Figure 2d* | Two-way ANOVA test | Group: p<0.0001, Df = 1, F = 58.20<br>Distance: p=0.97, Df = 15, F = 0.44<br>Interaction: p=0.33, df = 15, F = 1.13 | WT n = 26126 distances in eight mice; Hdh150 n = 58050 distances in six mice |
| *Figure 2e* | Two-way ANOVA test | p=0.35, Df = 1, F = 0.86 | WT rand n = 26126 distances in eight mice; Hdh150 rand n = 58050 distances in six mice |
| *Figure 3b* | Mann-Whitney test | p=0.031119 | WT n = 10 mice; Hdh150 n = 13 mice |
| *Figure 4a* | Mann-Whitney test | control vs 1 mM metformin p=0.084521, control vs 2.5 mM metformin p=0.023231 | Control n = 10, 1 mM metformin n = 11, 2.5 mM metformin n = 10. |
| *Figure 4b* | Mann-Whitney test | control siRNA vs MID1 siRNA p=0.008, control siRNA vs MID1 siRNA + metformin p=0.015 | Control siRNA n = 6, MID1 siRNA n = 6, MID1 siRNA + metformin n=6. |
| *Figure 4c* | RM two-way ANOVA | Treatment: p=0.0082, Df = 2, F = 5<br>Time: p<0.0001, Df = 47, F = 27.5<br>Interaction: p<0.0001, Df = 94, F = 5.9 | $n_{control}$ = 47, $n_{metformin\ 1mM}$ = 44, $n_{metformin\ 2.5mM}$ = 35 |
| *Figure 4d* | RM two-way ANOVA | Treatment: p=0.0021, Df = 3, F = 5.1<br>Time: p<0.0001, Df = 47, F = 64.1<br>Interaction: p<0.0001, Df = 141, F = 6.1 p<0.0001 | $n_{control}$ = 46, $n_{metformin}$ = 49, $n_{metformin+OA}$ = 51, $n_{OA}$ = 43 |
| *Figure 4f* | Unpaired t-test | p=0.0473 | Hdh150 n = 6; Hdh150 metformin n = 6 |
| *Figure 4h* | Unpaired t-test | p=0.0467 | Hdh150 n = 3; Hdh150 metformin n = 3 |

*Table 1 continued on next page*

Table 1 continued

| Figure | Test | Values | N |
|---|---|---|---|
| *Figure 4i* | Unpaired t-test | p=0.0062 | Hdh150 n = 3; Hdh150 metformin n = 3 |
| *Figure 4j* | Unpaired t-test | p=0.8766 | Hdh150 n = 3; Hdh150 metformin n = 3 |
| *Figure 5b* | Mann-Whitney test | WT vs Hdh150 p=0.023, Hdh150 vs Hdh150 met p=0.03, Hdh150 vs WT met p=0.012 | WT n = 1204 cells in eight mice; Hdh150 n = 933 cells in six mice; WT met n = 1915 cells in nine mice; Hdh150 met n = 1585 cells in six mice |
| *Figure 5c* | Mann-Whitney test | WT vs Hdh150 p=0.006; Hdh150 vs Hdh150 met p=0.007; Hdh150 vs WT met p=0.008 | WT n = 765 cells in eight mice; Hdh150 n = 695 cells in six mice; WT met n = 1199 in nine mice; Hdh150 met n = 1014 cells in six mice |
| *Figure 5d* | Two-way ANOVA test | Group: p<0.0001, Df = 3, F = 61.80<br>Time: p<0.0001, Df = 16, F = 345.9<br>Interaction: p<0.0001, Df = 48, F = 3.64 | WT n = 765 cells eight mice; Hdh150 n = 695 cells six mice; WT met n = 1199 cells nine mice; Hdh150 met n = 1012 cells six mice |
| *Figure 5e* | Chi-square test | p=0.62, df = 1; Chi-square = 0.24 | WT n = 765 cells eight mice; Hdh150 n = 695 cells six mice; WT met n = 1199 cells nine mice; Hdh150 met n = 1012 cells six mice |
| *Figure 5f* | Mann-Whitney test | WT vs Hdh150 p=0.03; Hdh150 vs Hdh150 met p=0.002; Hdh150 vs WT met p=0.003 | WT n = 765 cells eight mice; Hdh150 n = 695 cells six mice; WT met n = 1199 cells nine mice; Hdh150 met n = 1012 cells six mice |
| *Figure 5g* | Two-way ANOVA test | Group: p<0.0001, Df = 3, F = 85.96<br>Distance: p=0.99, Df = 45, F = 0.58<br>Interaction: p=0.0007, Df = 15, F = 2.63 | WT n = 765 cells eight mice; Hdh150 n = 695 cells six mice; WT met n = 1199 cells nine mice; Hdh150 met n = 1012 cells six mice |
| *Figure 5i* | Mann-Whitney test | WT vs Hdh150 p=0.002, Hdh150 vs Hdh150 Met p=0.002, Hdh150 vs. WT met p=0.02, WT vs Hdh150 Met p=0.82 | WT n = 10; Hdh150 n = 13; WT met n = 6; Hdh150 met n = 8 mice |
| Figures supplements | Test | values | n |
| *Figure 1—figure supplement 1e* | Mann-Whitney test | p=0.002 | n = 6 neurons, n = 6 astrocytes |
| *Figure 1—figure supplement 1f* | Mann-Whitney test | p=0.002 | n = 6 neurons, n = 6 astrocytes |

Table 1 continued on next page

*Table 1 continued*

| Figure | Test | Values | N |
|---|---|---|---|
| *Figure 1—figure supplement 3d* | Mann-Whitney test | In WT mice: SS vs SL p=0.5, SS vs SM p=0.9, SS vs LL p=0.1, SS vs LM p=0.2, SS vs MM p=0.1, SL vs SM p=0.4, SM vs LL p=0.1, SM vs MM p=0.1, LL vs MM p=0.9, LM vs MM p=0.4, LM vs SM, p=0.2.<br>In Hdh150 mice: SS vs SL p=0.8, SS vs SM p=0.5, SS vs SH p=0.9, SS vs SH p=0.9, SS vs LL p=0.9, SS vs LM p=0.9, SS vs LH p=1, SS vs MM p=0.1, SS vs MH p=0.1, SS vs HH p=0.4, SL vs SM p=0.8, SL vs SH p=0.9, SL vs LL p=0.9, SL vs LM p=0.9, SL vs LH p=0.8, SL vs MM p=0.5, SL vs MH p=0.6, SL vs HH p=0.7, SN vs SH p=0.6, SM vs LL p=0.7, SM vs LM p=0.6, SN vs LH p=0.4, SM vs MM p=0.3, SN vs MH p=0.4, SN vs HH p 0 0.6, SH vs LL p=0.7, SH vs LM p=1, SH vs LH p=1, SH vs MM p=0.4, SH vs MH p=0.3, SH vs HH p=0.6, LL vs LM p=0.9, LL vs LH p=1, LL vs MM p=0.5, LL vs MH p=0.7, LL vs HH p=0.8, LM vs LH p=0.6, LM vs MM p=0.3, LM vs MH p=0.3, LM vs HH p=0.5, LH vs MM p=0.3, LH vs MH p=0.3, LH vs HH p=0.5, MM vs MH p=0.7, NN vs HH p=0.8, MH vs HH p=0.9<br>In WT vs WT rand: WT mice: SS p=0.5, SL p=0.7, SM p=0.3, LL p=0.3, LM p=0.8, MM p=0.1.<br>In Hdh150 vs Hdh150 rand: SS p=0.6, SL p=1, SM p=1, SH p=0.6, LL p=0.7, LM p=0.6, LH p=0.2, MM p=0.6, MH p=0.8, HH p=0.7 NS | WT n = 72595 distances eight mice; Hdh150 n = 132009 distances six mice |
| *Figure 2—figure supplement 1a* | Mann-Whitney test | LL p=0.005; LM p<0.0001; MM p<0.0001 | WT n = 26126 Pearson's r in eight mice |
| *Figure 2—figure supplement 1b* | Mann-Whitney test | LL p=0.004; LM p=0.0006; LH p=0.041; MM p<0.0001; MH p=0.0002; HH p=0.01. In Hdh150 mice, compared to LL: MM p=0.049; MH p=0.005; HH p=0.009 | Hdh150 n = 58050 Pearson's r in six mice |
| *Figure 2—figure supplement 2c* | Mann-Whitney test | routine p=0.4, leak p=0.5, CI p=0.6, CI + CII p=0.5, ETS p=0.2 | WT n = 6 mice; Hdh150 n = 6 mice |
| *Figure 3—figure supplement 1b* | RM two-way ANOVA | Genotype: p=0.6, Df = 1, F = 0.3<br>Time: p<0.0001, Df = 6, F = 86.1<br>Interaction: p=0.6, Df = 6, F = 0.7 | WT n = 16 mice; Hdh150 n = 13 mice |
| *Figure 3—figure supplement 1c* | RM two-way ANOVA | Genotype: p=0.5, Df = 1, F = 0.5<br>Time: p<0.0001, Df = 9, F = 35.4<br>Interaction: p=0.03, Df = 9, F = 2.2 | WT n = 16 mice; Hdh150 n = 13 mice |
| *Figure 3—figure supplement 1d* | RM two-way ANOVA | Genotype: p=0.8, Df = 1, F = 3.4<br>Time: p<0.01, Df = 6, F = 3.4<br>Interaction: p=0.97, Df = 6, F = 0.2 | WT n = 16 mice; Hdh150 n = 13 mice |
| *Figure 3—figure supplement 1e* | Mann-Whitney test | p=0.3 | WT n = 10; Hdh150 n = 13; WT met n = 6; Hdh150 met n = 8 mice |
| *Figure 4—figure supplement 1a* | RM two-way ANOVA | Treatment p=0.0342, Df = 2, F = 3.5;<br>Time p<0.0001, Df = 47, F = 45.3; Interaction p<0.0001, Df = 94, F = 3.5 | $n_{control}$ = 36, $n_{metformin\ 1mM}$ = 42, $n_{metformin\ 2.5mM}$ = 44 |
| *Figure 4—figure supplement 1b* | RM two-way ANOVA | Treatment p=0.2986, Df = 1, F = 1.9;<br>Time p=0.0654, Df = 20, F = 1.8;<br>Interaction p=0.9988, Df = 20, F = 0.3. | control n = 7, metformin n = 8 |
| *Figure 4—figure supplement 1dc* | Unpaired t-test | p=0,1826 | Hdh150 n = 4; Hdh150 metformin n = 4 |
| *Figure 4—figure supplement 2a* | Mann-Whitney test | p<0.0001 | control n = 65, metformin n = 65 |
| *Figure 4—figure supplement 2b* | Mann-Whitney test | Q40 vs. Q40 Met p<0.0001 | Q40n = 43, Q40 Met n = 43 |
| *Figure 4—figure supplement 2c* | Mann-Whitney test | Ctrl vs. 5 mM p=0.0078, Ctrl vs. 10 mM p<0.0001 | n = 45 |
| *Figure 4—figure supplement 2d* | Mann-Whitney test | p<0.0001 | Control n = 72, arc-1 RNAi n = 74 |
| *Figure 4—figure supplement 2e* | Mann-Whitney test | p<0.0001 | Control n = 60, arc-1 RNAi n = 62 |
| *Figure 5—figure supplement 1b* | Unpaired t test, two-tailed | WT met vs. Hdh150 met p=0.39, WT vs. WT met p=0.7, Hdh150 vs. Hdh150 met p=0.9 | WT n = 11; Hdh150 n = 10; WT met n = 9; Hdh150 met n = 6 mice |

*Table 1 continued on next page*

*Table 1 continued*

| Figure | Test | Values | | N |
|--------|------|--------|---|---|
| *Figure 5—figure supplement 1c* | Unpaired t test, two-tailed | WT vs WT met p=0.024 | | WT n = 765 cells eight mice; Hdh150 n = 695 cells six mice; WT met n = 1199 cells nine mice; Hdh150 met n = 1012 cells six mice |
| *Figure 5—figure supplement 1d* | Unpaired t test, two-tailed | WT met: LL vs LM p=0.04 and LL vs MM p<0.0001; Hdh150 met: LL vs LM p=0.4, LL vs MM p=0.004 | | WT met n = 57140 Pearson's r in nine mice; Hdh150 met n = 49535 Pearson's r in six mice |
| *Figure 5—figure supplement 1e* | Mann-Whitney test | LM WT vs LM Hdh150; MM WT vs MM Hdh150 p=0.04 | | WT n = 26126 Pearson's r in eight mice; Hdh150 n = 58050 Pearson's r in six mice; WT met n = 57140 Pearson's r in nine mice; Hdh150 met n = 49535 Pearson's r in six mice |
| *Figure 5—figure supplement 1f* | Mann-Whitney test | SS p=0.1, SL p=0.1, SM p=0.1, LL p=0.4, LM p=0.3, MM p=0.2 | | WT met n = 140467 distances in nine mice; Hdh150 met n = 117485 distances in six mice |

DOI: https://doi.org/10.7554/eLife.38744.011

## Behavior changes in premanifest VFDO Huntington's disease mice

Next, we asked whether hyperactivity and increased synchronicity of cortical networks are associated with behavioral changes in the VFDO animals. A visual discrimination task did not show any aberrations in the Hdh150 animals (*Figure 3—figure supplement 1a–d*; *Figure 3—video 1*, *Figure 3—source data 1*, two-way ANOVA, not significant, see *Table 1* for p-values). In contrast, in an open-field test premanifest VDFO Hdh150 animals moved significantly more to the center than WT littermates suggesting anxiolytic effects of the VFDO changes (*Figure 3a,b*, *Figure 3—source data 1*, Mann-Whitney test, p=0.03). Distance travelled (as a measure of motility) did not differ between groups (*Figure 3—figure supplement 1e*, *Figure 3—source data 1*, Mann-Whitney test, not significant, see *Table 1* for p-values).

Taken together, we have found network hyperactivity in the cortex of VFDO Hdh150 mice combined with anxiolytic behavior.

## Metformin reduces mutant Htt protein load

Based on our previous observations that the MID1/PP2A/mTOR protein complex regulates the translation of mHtt protein (*Krauss et al., 2013*) and that treatment with metformin interferes with the MID1 complex (*Kickstein et al., 2010*), we hypothesized that metformin inhibits the MID1/PP2A/mTOR-mediated protein synthesis of mutant mHtt and is therefore a promising candidate molecule to reduce mHtt load and reverse symptoms associated with Huntington's disease.

In order to test for an effect of metformin on mHtt protein load and aggregation, HEKT cells stably expressing FLAG-tagged exon 1 of human mHTT carrying 83 CAG repeats were treated with 1 mM or 2.5 mM metformin, or with vehicle for 48 hr. Aggregation was quantified in a filter retardation assay. Metformin reduced the amount of aggregated FLAG-HTT in a concentration-dependent manner (*Figure 4a*, *Figure 4—source data 1*, Mann-Whitney test, control vs 2.5 mM metformin p=0.02).

To test whether the metformin effect on human exon 1 mHTT aggregates is mediated by a blockade of the MID1 protein complex, we depleted MID1 by siRNA-mediated knockdown in the cell line expressing FLAG-HTT exon 1 with 83 CAG repeats, in presence or absence of metformin. While depletion of MID1 reduced mHTT aggregation, no additive effect of metformin treatment on mHTT

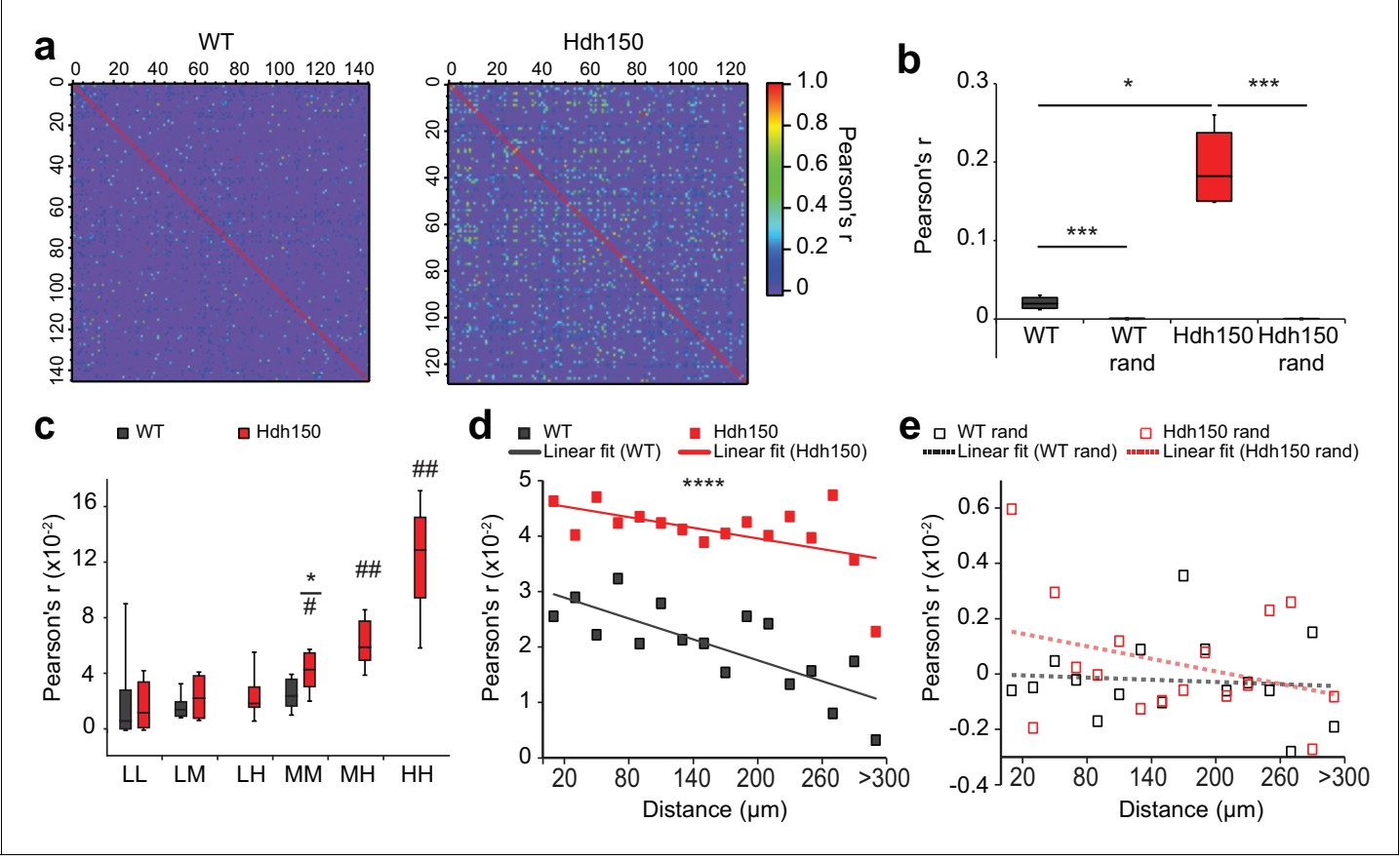

**Figure 2.** Presymptomatic Hdh150 mice exhibit an increased synchronicity of cortical microcircuits. (**a**) Color-coded Pearson's r matrices calculated from representative recordings of WT (left) and Hdh150 (right) mice. Silent cells were excluded from the analysis. Right, color-coded scale of Pearson's r values. (**b**) Overall Pearson's correlation coefficient (Pearson's r) in WT (dark grey) and Hdh150 (red) mice for experimental (filled) and randomized (open) raster data. Mann-Whitney test, WT vs. Hdh150 p<0.05; WT vs. WT rand p<0.0001; Hdh150 vs. Hdh150 rand p<0.0001 (**c**) Pearson's r for combinations of neuronal pairs (LL: low-low, LM: low-medium, LH: low-hyper, MM: medium-medium, MH: medium-hyper, HH: hyper-hyper) in WT (dark grey) and Hdh150 (red) mice. * pairwise comparisons between a pair of WT and Hdh150 mice. # comparisons of functional subgroup pairs to the low-low pair within the same genotype. The pairs involving hyperactive neurons could only be analyzed in Hdh150 mice. Mann-Whitney test, WT vs. Hdh150 mice: MM p<0.05 in Hdh150 mice; compared to LL: MM p<0.05, MH p<0.01, HH p<0.01 (**d,e**) Relationship between Pearson's r and distance between neuronal pairs in WT (black) and Hdh150 (red) mice (**d**) and randomized data (**e**). Lines represent the linear fit of WT and Hdh150 experimental data. Two-way ANOVA (**d**) Genotype: p<0.0001 Distance: p=0.97, Interaction: p=0.3, (**e**) Genotype = 0.35, p=0.3, Interaction: p=0.8.
DOI: https://doi.org/10.7554/eLife.38744.012

The following source data and figure supplements are available for figure 2:

**Source data 1.** Numerical values of *Figure 2* and associated supplement figures.
DOI: https://doi.org/10.7554/eLife.38744.015
**Figure supplement 1.** Randomization of experimental data to assess specific network synchronicity.
DOI: https://doi.org/10.7554/eLife.38744.013
**Figure supplement 2.** Presymptomatic Hdh150 mice did not exhibit alteration of mitochondria respiration.
DOI: https://doi.org/10.7554/eLife.38744.014

aggregates was observed, suggesting that MID1 and metformin indeed act through the same pathway (*Figure 4b*, *Figure 4—source data 1*, Mann-Whitney test, control siRNA vs MID1 siRNA p=0.009; control siRNA vs MID1 siRNA +metformin p=0.02).

We had shown previously that the MID1/PP2A/mTOR protein complex regulates the translation efficiency of the human *HTT* mRNA in a repeat-dependent manner (*Krauss et al., 2013*). We therefore looked at a possible influence of metformin on the protein synthesis rate of mHTT exon one protein using a previously described FRAP (Fluorescence recovery after photo bleaching) - based assay that allows monitoring of protein translation rates in living cells (*Krauss et al., 2013*). We

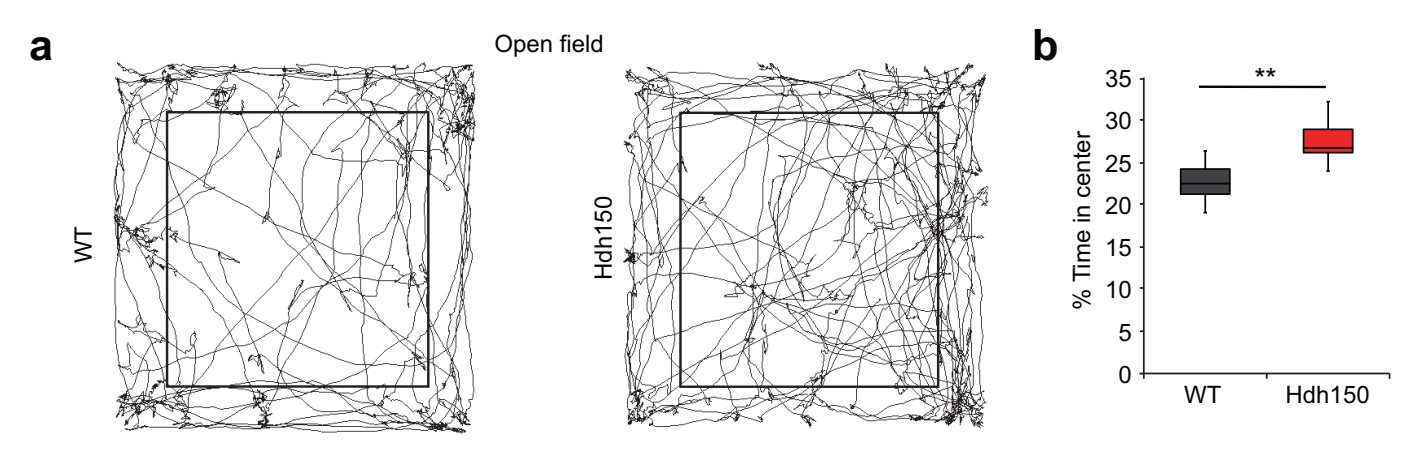

**Figure 3.** Presymptomatic VFDO Hdh150 mice exhibit anxiolytic behavior. (a) Representative travel pathways of WT (left) and presymptomatic Hdh150 (right) mice analyzed in a 5 min open field test. (b) Increased explorative behavior of Hdh150 animals compared to the WT mice. Mann-Whitney test, p<0.05.

DOI: https://doi.org/10.7554/eLife.38744.016

The following video, source data, and figure supplement are available for figure 3:

**Source data 1.** Numerical values of *Figure 3* and associated supplement figures.
DOI: https://doi.org/10.7554/eLife.38744.018

**Figure supplement 1.** Presymptomatic VFDO Hdh150 mice did not exhibit deficit in visual discrimination test and explorative behavior in novel object recognition test.
DOI: https://doi.org/10.7554/eLife.38744.017

**Figure 3—video 1.** Example of visual discrimination task performed by a trained mouse, real time.
DOI: https://doi.org/10.7554/eLife.38744.019

detected a clear reduction in the protein synthesis rate of a GFP-Htt fusion protein carrying 49 repeats in the metformin-treated samples in a concentration-dependent manner in primary neurons (*Figure 4c*, *Figure 4—source data 1*, RM two-way ANOVA p=0.008). This effect was confirmed in N2A cells (*Figure 4—figure supplement 1a*, *Figure 4—source data 1*, RM two-way ANOVA p=0.03). To further support the contribution of the MID1/PP2A/mTOR protein complex and PP2A activity to this effect, the GFP-Htt transfected cells were subsequently either (i) mock treated, (ii) treated with only metformin, (iii) treated with ocadaic acid (OA), or (iv) co-treated with metformin and OA. OA is an inhibitor of PP2A activity. As expected, OA significantly increased translation rates of the *GFP*-Htt reporter mRNA and metformin did not have a reducing effect on the translation rates in cells co-treated with OA suggesting that indeed the metformin effect is mediated by PP2A activity (*Figure 4d*, *Figure 4—source data 1*, RM two-way ANOVA p=0.002).

To analyze metformin effects on early signs of pathology *in vivo*, VFDO Hdh150 mice were fed with, or without 5 mg/ml metformin in the drinking water. Metformin did not significantly reduce drinking volume (*Figure 4—figure supplement 1b*, *Figure 4—source data 1*, RM two-way ANOVA not significant, see *Table 1* for p-values). After 3 weeks of treatment, we looked at phosphorylation patterns of the PP2A/mTOR target S6 and the amount of mHtt protein relative to wild-type Htt in whole brain tissue. The metformin-treated group showed a significant reduction of S6 phosphorylation suggesting an increase in PP2A activity (*Figure 4e and f*, *Figure 4—source data 1*, unpaired t-test p=0.05). At the same time, a slight tendency (not significant) of reduced mHtt was detected suggesting that metformin has an influence on mHtt expression (*Figure 4—figure supplement 1c and d*, *Figure 4—source data 1*, unpaired t-test not significant, see *Table 1* for p-values). A significant reduction of mHtt expression, however, became clearly visible after 11 weeks of treatment in cortical tissue (*Figure 4g–j*, *Figure 4—source data 1*, unpaired t-test, mHtt/wtHtt p=0.05, mHtt/Gapdh p=0.002, wtHtt/Gapdh p=0.9).

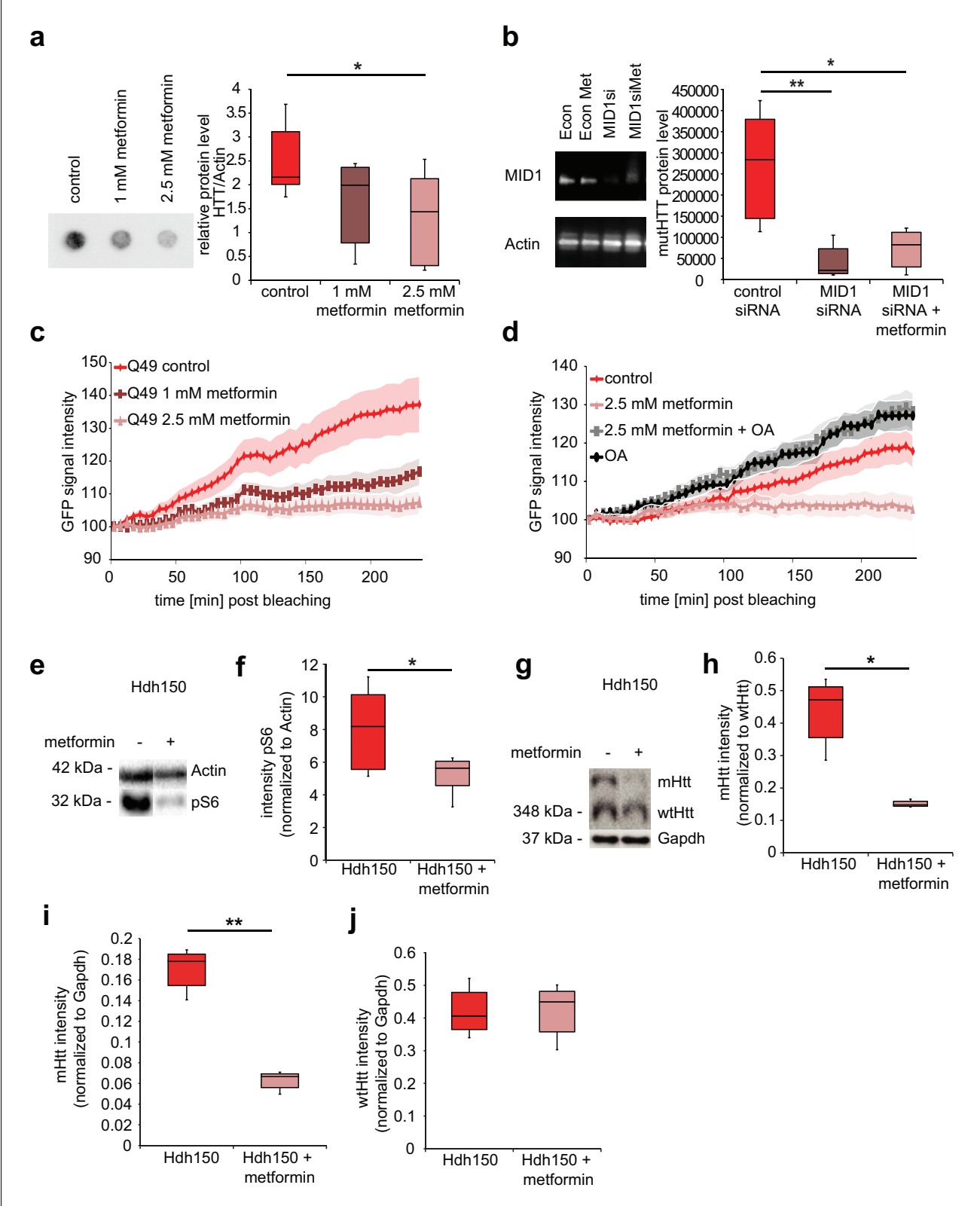

**Figure 4.** Metformin reduces translation rates of mutant HTT through MID1/PP2A protein complex *in vitro* and decreases both S6 phosphorylation and mutant Htt protein load in Hdh150 animals. (a) FLAG-HTT detected on a filter retardation assay after treatment with and without 1 mM and 2.5 mM metformin. Quantification on right panel. Mann-Whitney test, control vs. 1 mM metformin p=0.08; control vs. 2.5 mM metformin, p<0.05. (b) Stable cell line expressing FLAG-HTT exon1 with 83 CAG repeats transfected with *MID1*-specific siRNAs or control siRNAs in the presence or absence of 2.5 mM

*Figure 4 continued on next page*

*Figure 4 continued*

metformin. FLAG-HTT detected on a filter retardation assay. Efficiency of the knock-down including Actin as a loading control is shown on a western blot (left panel). Quantification of filter retardation assay on right panel. Mann-Whitney test, control siRNA vs. MID1 siRNA p<0.01; control siRNA vs. MID1 siRNA + metformin p<0.05. (**c**) Protein translation rate of GFP-tagged mutant Htt exon1 (49 CAG repeats) in primary cortical neurons measured in a FRAP-based assay, over a time frame of 4 hr. Lines show the GFP-signal intensity over time in mock-treated (control) and metformin-treated (1 mM and 2.5 mM) cells. Lines represent means, shadowed areas standard deviations. Repeated measures two-way ANOVA, treatment p<0.01, time p<0.0001; interaction p<0.0001. (**d**) Protein translation rate measured in a FRAP-based assay (see c). Lines show the GFP-signal intensity over time in mock-treated (control), metformin-treated (2.5 mM), ocadaic acid (OA)-treated and metformin +OA-treated cells. Shadowed areas show SEM. Repeated measures two-way ANOVA, treatment p<0.01, time p<0.0001, interaction p<0.0001. (**e**) Transgenic Hdh150 mice received metformin-containing water (5 mg/ml, Hdh150 +metformin) or pure water (Hdh150) over a period of 3 weeks. Whole brain lysates were analyzed for the phosphorylation of S6, the expression of total S6, mHtt and wtHtt on western blots. Representative western blots are shown. (**f**) Quantification of pS6 relative to S6. Unpaired t-test, p<0.05. (**g**) mHtt and wt Htt proteins of prefrontal cortex lysates analyzed on western blots after 11 weeks of treatment with metformin (5 mg/ml, Hdh150 +metformin) or pure water (Hdh150). Representative western blots are shown. (**h**) Quantification of mHtt relative to wtHtt. Treatment of 5 mg/ml metformin in the drinking water showed a significant reduction of mHtt protein compared to water control treatment. Unpaired t-test p <<0.05 (**i**) Quantification of mHtt relative to Gapdh. Unpaired t-test, p<0.01. (**j**) Quantification of wtHtt relative to Gapdh. Unpaired t-test, p=0.88.
DOI: https://doi.org/10.7554/eLife.38744.020

The following source data and figure supplements are available for figure 4:

**Source data 1.** Numerical values of *Figure 4* and associated supplement figures.
DOI: https://doi.org/10.7554/eLife.38744.023
**Figure supplement 1.** Metformin reduces mutant Htt protein translation and does not change drinking behavior of Hdh150CAG animals.
DOI: https://doi.org/10.7554/eLife.38744.021
**Figure supplement 2.** Metformin treatment rescues motility impairment in a *C.elegans* model.
DOI: https://doi.org/10.7554/eLife.38744.022

## Metformin reverses signs of Huntington's disease pathology

Our data suggest metformin as a promising molecule to interfere with VFDO Huntington's disease biochemical and cellular pathological changes. We initially used a *C. elegans* model of polyQ-mediated diseases to test whether metformin effectively ameliorates disease symptoms in an easily controllable model with short lifespan. The *C. elegans* worms carry a transgene encoding YFP-tagged Q40 polypeptide in body wall muscle cells (*Morley et al., 2002*). Adult Q40::YFP nematodes exhibit intracellular aggregates of polyglutamine-containing protein and develop progressive paralysis over time, which is reflected in significantly reduced motility. We counted aggregates in metformin-treated and untreated nematodes. Moreover, we assessed their motility in a liquid thrashing experiment, in which worms are placed in liquid and the frequency of lateral swimming (thrashing) movement is analyzed as a measure of motility. We found that 5 days of metformin treatment reduced the number of intracellular inclusion bodies significantly and rescued motility impairment (*Figure 4— figure supplement 2a and b*, *Figure 4—source data 1*, Mann-Whitney, p<0.0001). Since bacteria can metabolize metformin (*Cabreiro et al., 2013*), we confirmed the results on heat-inactivated OP50 bacteria (*Figure 4—figure supplement 2c*, *Figure 4—source data 1*, Mann-Whitney test, control vs 5 mM metformin p=0.008, control vs 10 mM metformin p<0.0001). siRNA-mediated knock-down of arc-1, the *C.elegans* MID1 homolog, leads to a reduction of inclusion bodies and improved motility similar to the metformin effects (*Figure 4—figure supplement 2d and e*, *Figure 4—source data 1*, Mann-Whitney test, p<0.0001) and confirms that the MID1/PP2A/mTOR protein complex underlies metformin effects.

We then asked whether metformin could rescue the altered cortical activity *in vivo* in Hdh150 mice. Metformin treatment did not affect the density of OGB-1 stained cells (*Figure 5—figure supplement 1a,b*, *Figure 5—source data 1*, WT: 1413 ± 74 cells/mm$^2$ (n = 11 mice), Hdh150: 1456 ± 90 cells/mm$^2$ (n = 10 mice), WT met: 1351 ± 65 cells/mm$^2$ (n = 9 mice) and Hdh150 met: 1448 ± 60 cells/ mm$^2$ (n = 6 mice), unpaired t-test, not significant, see *Table 1* for p-values). Notably, 3 weeks of metformin treatment in the drinking water completely restored the proportion of active cells (*Figure 5b*, *Figure 5—source data 1*, Hdh150: 77.2 ± 3.5% (n = 6 mice), Hdh150 met: 64.4 ± 4.1% (n = 6 mice), Mann-Whitney test p<0.05) and the average frequency of Ca$^{2+}$ transients (*Figure 5a,c*, *Figure 5—source data 1*, Hdh150: 1.2 ± 0.1 trans/min (n = 6 mice), Hdh150 met: 0.7 ± 0.06% (n = 6 mice), Mann-Whitney test p<0.05). The individual traces of treated Hdh150 animals were indistinguishable from untreated WT animals (*Figure 5a*, *Figure 5—figure supplement 1c*).

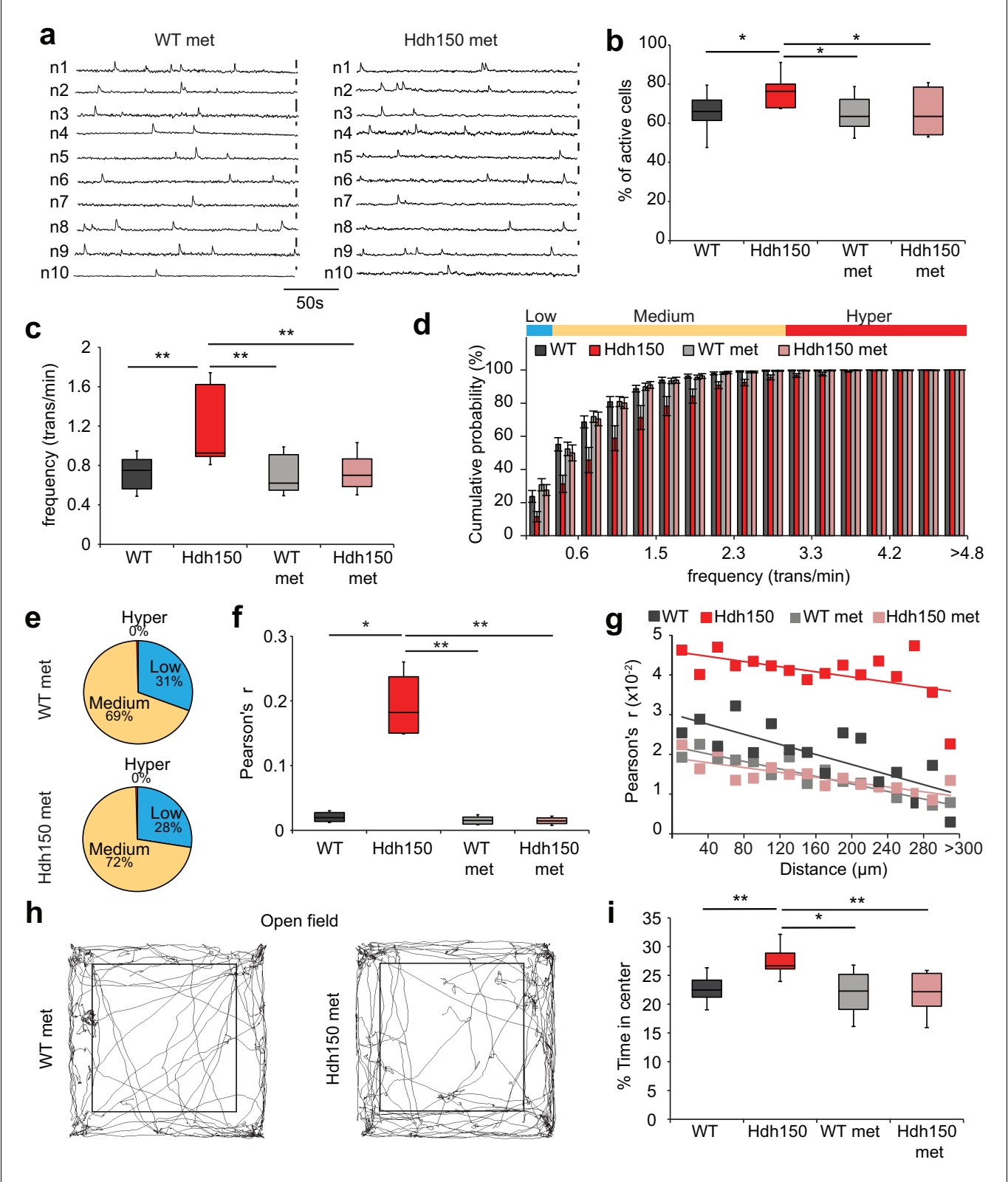

**Figure 5.** Metformin treatment reverses pathological neuronal network activity and behavioral abnormalities in presymptomatic VFDO Hdh150 mice. (a) Representative traces of spontaneous Ca²⁺ transients of 10 neurons recorded *in vivo* in WT and Hdh150 mice after metformin treatment. Vertical scale bar: 40% dF/F. (b) Relative proportion of spontaneously active neurons in WT (dark grey), Hdh150 (red), WT metformin-treated (light grey) and Hdh150 metformin-treated (light red) mice. Mann-Whitney test, WT vs. Hdh150, p<0.05; Hdh150 vs. Hdh150 met, p<0.05; Hdh150 vs. WT met, p<0.05.

*Figure 5 continued on next page*

*Figure 5 continued*

(c) Significant reduction in the spontaneous Ca$^{2+}$ transient frequency to WT levels in Hdh150 mice after metformin treatment (red vs. light red). Mann-Whitney test, WT vs. Hdh150, p<0.01; Hdh150 vs. Hdh150 met, p<0.01; Hdh150 vs. WT met, p<0.01. (d) Cumulative frequency distributions of Ca$^{2+}$ transients in WT (dark grey), Hdh150 (red), metformin-treated WT (light grey) and metformin-treated Hdh150 (light red) mice. Top, color-coding of active neurons by frequency. Two-way ANOVA test, Group: p<0.0001; Time: p<0.0001; Interaction: p<0.0001. (e) Pie charts showing the relative proportion of low (blue), medium (orange) and hyperactive (red) neurons in layer 2/3 of the visual cortex in WT (top) and Hdh150 (bottom) mice after metformin treatment. Chi-square test, p=0.62, Chi-square = 0.24. (f) Comparison of Pearson's r between a pair of neurons in WT (dark grey), Hdh150 (red), metformin-treated WT (light grey) and metformin-treated Hdh150 (light red) mice. Mann-Whitney test, WT vs. Hdh150, p<0.05, Hdh150 vs. Hdh150 met, p<0.01, Hdh150 vs. WT met, p<0.01. (g) Relationship between pairwise Pearson's r and pairwise distance in metformin-treated WT (light grey) and Hdh150 (light red) mice. Two-way ANOVA test, group p<0.0001; Distance p=0.09; Interaction p<0.001. (h) Representative travel pathways of a metformin-treated WT (left) and pre-symptomatic Hdh150 (right) mice analyzed in a 5 min open-field test. (i) Decrease in the explorative behavior of metformin-treated Hdh150 animals. Mann-Whitney test, WT vs. Hdh150, p<0.01; Hdh150 vs. Hdh150 met, p<0.001; Hdh150 vs. WT met, p<0.05; WT vs. Hdh150 met, p=0.8.

DOI: https://doi.org/10.7554/eLife.38744.024

The following source data and figure supplement are available for figure 5:

**Source data 1.** Numerical values of *Figure 5* and associated supplement figures.

DOI: https://doi.org/10.7554/eLife.38744.026

**Figure supplement 1.** Metformin treatment does not affect cell density or Ca$^{2+}$ transient dynamics

DOI: https://doi.org/10.7554/eLife.38744.025

Importantly, in these experiments, metformin acts specific on dysregulated network components. Only the AUC of calcium transients was slightly yet significantly affected in WT mice which might be due to an increase of baseline calcium concentration induced by the activation of MID1/PP2A/mTOR signaling pathway (*Figure 5—figure supplement 1c*, *Figure 5—source data 1*). One of the hallmarks of the VFDO Hdh150 mice was the distinct functional subgroup of hyperactive neurons (*Figure 1*). We thus assessed the effect of metformin treatment on the relative proportions of low, normal and hyperactive subgroups. Treatment with metformin in Hdh150 mice led to the complete abolishment of the hyperactive subgroup (*Figure 5d*, *Figure 5—source data 1*, two-way ANOVA p<0.0001 and *Figure 5e*, *Figure 5—source data 1*, Chi-square test not significant, see *Table 1* for p-values) and restored the relative proportion of functional subgroups. The spatial distribution of functional subgroups remained unchanged by metformin treatment (*Figure 5—figure supplement 1f*, *Figure 5—source data 1*, Mann-Whitney test, not significant, see *Table 1* for p-values). This complete restoration of dysregulated microcircuit activity was also evident in the cumulative frequency distribution (*Figure 5d*, *Figure 5—source data 1*) and all other network measures found to be aberrant in young Hdh150 animals, such as synchronicity (*Figure 5f and g*, *Figure 5—figure supplement 1d–e*, *Figure 5—source data 1*).

Accompanying the network dysfunction, we observed anxiolytic behavior in the VFDO Hdh150 mice (see *Figure 3a–b*). Treatment with metformin fully reversed the anxiolytic behavior (*Figure 5h and i*, *Figure 5—source data 1*, Mann-Whitney test, WT vs Hdh150 p=0.002, Hdh150 vs Hdh150 met p=0.002, Hdh150 vs WT met p=0.03, WT vs Hdh150 met p=0.8).

In conclusion, our data suggest that metformin, by interfering with the translation rate of mHtt protein, reduces protein load in cell culture and *in vivo* and reverses early Huntington's disease-related network dysregulations as well as anxiety-related behavioral aberrations in mice. Furthermore, usage of metformin in adult *C. elegans,* a model of polyQ disease, also significantly influences inclusion bodies formation and motility.

## Discussion

Here, we have identified a dysregulation of spontaneous neuronal activity in the visual cortex in a very early stage of a mouse model of Huntington's disease that corresponds to a very early stage in the premanifest stage in Huntington's disease mutation carriers (*Figure 1a,b*), which might indeed be a distinct pathological disease stage which is very far from disease onset (VFDO). Correspondingly, the visual cortex is one of the first structures affected in patients (*Labuschagne et al., 2016*; *Dogan et al., 2013*). This dysregulation is characterized by an

increase in cortical network activity patterns, the emergence of a functional subgroup of hyperactive neurons, and enhanced synchronicity. Overall visual cortex functioning seems to be preserved at this early time point of Huntington's disease course. Network changes are accompanied by subtle behavior alterations including an anxiolytic phenotype, suggesting that at least part of the brain-wide circuitry exhausted its compensational reserve. Anxiety-related abnormalities including anxiolytic behavior have previously been described in the preclinical phase in several other rodent Huntington's disease models ((*Nguyen et al., 2006*), reviewed in [*Pouladi et al., 2013*]). So far, changes described here represent the earliest identified abnormalities in cortical pathophysiology and behavior in heterozygous Hdh150 animals, which closely resemble the human disease (*Lin et al., 2001*; *Heng et al., 2007*; *Brooks et al., 2012*; *Tallaksen-Greene et al., 2005*). Furthermore, we show that the type II diabetes drug metformin inhibits the translation of mHtt protein and thereby decreases mHtt protein load *in vitro* and *in vivo*. Promisingly, this leads to a complete restoration of VFDO network activity patterns as well as behavior abnormalities under chronic metformin therapy.

Our data report primal changes in cortical network function in the VFDO stage of Huntington's disease. A recognition of the network as a pathophysiological entity has recently been suggested in the context of Alzheimer's disease (*Busche et al., 2008*; *Iaccarino et al., 2016*). Moreover, focus has shifted away from the mechanisms accompanying neuronal and network degeneration and instead moved toward small and subtle functional changes at very early stages of the disease when irrevocable damage to the network has not yet occurred (*Busche and Konnerth, 2016*). Indeed, hyperactive neurons are associated with both advanced and early stages of Alzheimer's disease, independent of plaque formation (*Busche et al., 2012*; *Busche et al., 2008*). In addition, evidence points toward hyperactive neurons preventing the cortex-wide propagation of slow oscillations in early Alzheimer's disease (*Busche et al., 2015*).

We here describe a similarly distinct hyperactive phenotype in very early stages of Huntington's disease. We conclude that neuronal hyperactivity may be a principle mechanism that develops early not only in Alzheimer's disease but also in other neurodegenerative diseases. Thus, the notion of early network dysregulation as a therapeutic target may have broad implications.

Similar to early stages of Alzheimer's disease hyperactive neurons in Hdh150 animals emerge in the absence of aggregate formation. Also hyperactive cells do not cluster, and reactive astrocytes or cells with activated caspase-3 as a marker of early apoptosis are not found in these early stages of Huntington's disease. Furthermore, cortical neurons did not exhibit metabolic dysregulation as measured in a mitochondrial respiration assay. Therefore, we may postulate that the cortex merely responds to early pathophysiological events already commencing in subcortical regions, e.g. the striatum. This is well in line with the current emerging hypotheses of disease progression in Alzheimer's disease. Young Alzheimer's disease animals develop hyperactivity in a plaque-independent fashion first in the hippocampus (possibly due to higher vulnerability of the hippocampus in Alzheimer's disease), followed by a similar hyperactivity pattern in the cortex later in the disease process (*Busche et al., 2012*; *Busche et al., 2008*). Furthermore, in Alzheimer's disease patients, degeneration of cortical projection targets of the hippocampus is associated with hippocampal hyperactivity indicating a connectivity-based spread of network dysregulations eventually leading to neurodegeneration (*Putcha et al., 2011*). Our data indeed suggest an altered activity in subcortical drivers, since unspecific alteration of excitability in individual neurons is unlikely to lead to an increase in synchronicity, but rather would result in a random increase in firing.

Our data was collected at very early stages of the disease in the Hdh150 mouse model, which corresponds to the VFDO stage in Huntington's disease patients. The importance of expression of mutant Htt protein during very early phases for disease development has been demonstrated in the BACHD:CAG-Cre$^{ERT2}$ mouse (*Molero et al., 2016*). With the help of tamoxifen treatment expression of mutant Htt was turned off early postnatally. Still the typical symptoms of Huntington's disease at three and nine months of age were observed in the animals. We conclude that only at the VFDO stages, when cellular and network degeneration have not yet been established, preventive strategies will be most effective; only then can we still rescue small homeostatic shifts, prevent spreading and potentially stabilize network function.

Phenotype reversal could be demonstrated in a tetracyclin-dependent conditional mouse model for Huntington's disease. Both neuropathological findings and behavior aberrations were found to disappear when mHtt protein production was stopped through a tet-off regulation mechanism in the

adult animal (*Yamamoto et al., 2000*). Additional support for the beneficial effect of suppression of aberrant protein production on the Huntington's disease phenotype stems from several studies with RNA interference (siRNA), or antisense oligonucleotides showing that gene suppression reducing mHtt protein load by 40% or more, is sufficient to significantly ameliorate the Huntington's disease phenotype (*HD iPSC Consortium, 2017*; *Harper et al., 2005*; *Keiser et al., 2016*; *Stanek et al., 2014*). These studies have demonstrated, that (i) the earlier suppression takes place the more robust and beneficial effects on behavior phenotypes are [reviewed in (*Keiser et al., 2016*)] and (ii) that even transient suppression of Htt protein during early disease stages was sufficient to obtain long-term effects on the disease phenotype lasting for months, far beyond the treatment period (*Lu and Yang, 2012*; *Kordasiewicz et al., 2012*).

However, developing siRNA and antisense oligonucleotides technologies into therapeutics for clinical use is difficult and a long way to go. Difficulties here include toxicity and modes of delivery: so far oligonucleotides have to be regularly injected into the cerebrospinal fluid, which is a huge effort for patients and physicians. Furthermore, a short N-terminal fragment of the mHtt protein, mHttexp1p, that is produced by incomplete exon one splicing and a short poly-adenylated mRNA in several animal models as well as in Huntington's disease patients rather than full-length mHtt protein was found to be particularly pathogenic. Its occurrence correlates well with age of onset and severity of the disease. This short mRNA is difficult to target by oligonucleotide strategies (*Neueder et al., 2017*; *Sathasivam et al., 2013*). We show here that a well-known, widely used, orally delivered small compound, metformin, suppresses mHtt production by targeting both, full-length and mHttex1p, *in vitro* and *in vivo* and thereby significantly reduces mHtt protein load, which makes it a very promising candidate for chronic early onset Huntington's disease therapy.

Metformin is an FDA-approved, inexpensive biguanide that has been used in patients with Type II diabetes for decades and is under discussion for cancer preventive therapy (*Demir et al., 2014*; *Micic et al., 2011*). Very recently, metformin has been shown to rescue core phenotypic features in a mouse model for fragile X-syndrome, a neurodevelopmental disorder, by normalizing ERK signalling (*Gantois et al., 2017*).

Metformin had been brought in as a promising compound in Huntington's disease previously. It has been found to protect cells from the toxicity of mutant Huntingtin protein in a cell culture model (*Jin et al., 2016*). In a study on R6/2 animals, a very aggressive model for Huntington's disease, Ma and colleagues had found a significant effect on survival rates and hind clasping in male animals only when given metformin in the drinking water starting from week 5 (*Ma et al., 2007*). In relation to the phenotype in the R6/2 animals that develops severe aberrations from 4 weeks on this is a late time point and would be placed in the motor phase stage when projected to the phenotypic time line given in *Figure 1a*. We hypothesize here that preventive treatment at a very early stage is important to substantially and stably influence the disease. This is in agreement with observations in a mouse model for spinocerebellar ataxia I, another neurodegenerative disease based on CAG expansion and studies with antisense oligonucleotides in a Huntington's disease model. Both studies show that gene suppression has more stable effects on the phenotype when performed early enough (*Kordasiewicz et al., 2012*; *Rubinsztein and Orr, 2016*; *Zu et al., 2004*). In line with that only two phenotypic features had been found to react on late metformin treatment- survival rates and frequency of clasping- in the R6/2 animals. Also and again as expected, effect size on animal survival was quite small (p=0.02). Gender differences in response rate seen in this study can possibly be explained by gender differences in disease development and progression at the motor stage, which had been observed in several mouse models (*Menalled et al., 2009*). When disease progression differs, differences in blood brain barrier permeability can be expected (reviewed in [*Sweeney et al., 2018*]) which then is likely to lead to gender specific variation in bioavailability of metformin in the brain. In contrast to this study we here show that *in vivo* metformin has highly significant effects (p=0.03 to p<0.0001) already on primal changes in the very early, VFDO phases of Huntington's disease making metformin a promising compound for the development of a therapeutic scheme that is based on early prevention of pathology development. While we focused on male mice in this study, to reduce physiological variability due to hormone fluctuations, at the VFDO stage brain barrier changes are not expected to influence bioavailability of metformin.

Metformin was suggested to lead, through AMPK activation, to a reduction of mHtt aggregates *in vitro* (*Walter et al., 2016*; *Vázquez-Manrique et al., 2016*). In our previous work, however, we

have shown that in cortical neurons metformin does not induce phosphorylation of the AMPK target ACC at all and only when given chronically it induces phosphorylation of AMPK itself in vitro. When giving 5 mg/ml metformin in the drinking water for 16 – 24 days to wildtype animals, while phosphorylation of S6 is significantly reduced, AMPK phosphorylation does not change in whole brain extracts (*Kickstein et al., 2010*). This indicates that AMPK activation is likely to depend on the dose. In WT animals as in preclinical Huntington's disease animals the blood-brain barrier is intact, which limits bioavailability of metformin in the brain. Metformin concentrations needed to influence mTOR/PP2A activity seem to be significantly lower than those needed to influence AMPK activity. Like in the Kickstein et al. paper, in the present study, we used 5 mg/ml metformin in the drinking water, a concentration at which AMPK activation is not expected, but as we show here in the Hdh150 animals, metformin has a significant effect on the phosphorylation of the mTOR/PP2A target S6.

The effect of Htt loss on brain function is still under debate. SiRNA studies suggest that postnatal reduction of endogenous Htt protein is well tolerated (summarized in [*Keiser et al., 2016*]). However, conditional knock-out animals with a perinatal loss of around 40% of Htt protein in the forebrain show a neurodegenerative phenotype (*Dragatsis et al., 2000*). Likewise, depletion of Htt protein in the adult brain leads to progressive behavior deficits (*Dietrich et al., 2017*). We demonstrate here that metformin has a very specific effect on the expression of mHtt protein only, leaving wild-type Htt that is produced from the non-mutated allele in dominant Huntington's disease untouched (*Figure 4*). This makes metformin the only compound available at present with a specific effect on only mutant but not wild-type Htt protein.

In support of an effect of metformin in Huntington's disease patients an *in silico* comparison of cognitive performance of Huntington's disease patients treated with metformin was performed. In this study, using the Enroll patient cohort, it was shown that diabetic Huntington's disease patients in the manifest stage treated with metformin had a better cognitive performance than Huntington's disease patients not treated with metformin (*Hervás et al., 2017*).

Our data indicate that metformin treatment reverses all cortical network dysregulations *in vivo* in the premanifest VFDO Hdh150 mice including functional sub-group distribution, frequency and synchronicity. Regaining network stability shows promise for ameliorating the molecular pathophysiology, probably by activating intrinsic repair mechanisms as shown in the context of Alzheimer's disease (*Iaccarino et al., 2016*; *Keskin et al., 2017*). Following this network-centered view, restoration of network functions might also prevent secondary damage to the neuronal microcircuit. We therefore propose a shift in experimental treatment strategies: rather than exploring single pathways for target, we might also consider re-balancing network function in the VFDO stages of the disease.

Taken together, our data provide evidence for the existence of a pathophysiological entity very far from onset of the manifest disease (VFDO) characterized by early homeostatic changes of network activity and, associated with that, subtle behavior alterations. The data also strongly support the observation that, similar to humans, the disease in mice develops over a long period of time. This provides an early critical window of vulnerability and gives opportunities for early therapeutic interference with disease development. So far, all attempts to develop a causative therapy for Huntington's disease have been unsuccessful [summarized in (*Crook and Housman, 2011*; *Clabough, 2013*)]. In terms of therapeutic intervention, consideration should be given to a chronic treatment of mutation carriers, which covers the critical windows of vulnerability, as early as in the VFDO stages. Such a strategy avoids delaying intervention until clinical signs of the disease are evident, implying that substantial brain damage has already occurred. Our data suggest that metformin has the potential to reduce mHtt protein load and to substantially influence the early development of pathology and, as seen in a *C. elegans* model, protein aggregation and movement aberrations which are pathognomonic for later disease stages. It is an inexpensive substance, well known in long-term clinical usage and has a defined, relatively benign spectrum of side effects. Prescription to mutation carriers from young adulthood on (or even earlier) is possible and will cover these newly discovered critical windows of opportunity for therapy.

# Materials and methods

**Key resources table**

| Reagent type (species) or resource | Designation | Source or reference | Identifiers | Additional information |
|---|---|---|---|---|
| Genetic reagent (*C.elegans*) | *C.elegans* strain AM141, genotype rmIs133 | University of Minnesota | AM141 (WormBase ID) RRID:WB-STRAIN:AM141 | |
| Genetic reagent (*M. Musculus*) | Hdh150 | Jackson Laboratory | #004595 RRID:IMSR_JAX:004595 | Only males were used |
| Cell line (*H. sapiens*) | HEK 293T/17 | *Scherzinger et al. (1997)* | CRL-11268 RRID:CVCL_1926 | |
| Cell line (*M. Musculus*) | Neuro-2A | ATCC | ATCC CCL131 RRID:CVCL_0470 | |
| Cell line (*M. Musculus*) | primary cortical neurons | isolated from NMRI (Janvier) | | |
| Transfected construct | pEGFP-C1-Httex1 | *Krauss et al., 2013* | Self-cloned | |
| Antibody | rabbit anti-activated-caspase-3 | Cell signaling | 9661 RRID:AB_2341188 | 1 to 500 |
| Antibody | mouse anti-NeuN | Millipore | MAB377 RRID:AB_2298772 | 1 to 500 |
| Antibody | rabbit anti-GFAP | Dako | Z0334 RRID:AB_10013382 | 1 to 1500 |
| Antibody | rabbit anti-Htt | Abcam | ab109115 RRID:AB_10863082 | WB: 1:850, IHC: 1:200 |
| Antibody | Alexa 546 goat anti-rabbit | Invitrogen | A11035 RRID:AB_143051 | 1 to 300 |
| Antibody | Alexa 647 goat anti-mouse | Invitrogen | A21235 RRID:AB_141693 | 1 to 300 |
| Antibody | Cy2 donkey anti-rabbit | Jackson Immuno Research | 711-225-152 RRID:AB_2340612 | 1 to 300 |
| Antibody | Alexa 488 goat anti-rabbit | Life technologies | A11008 RRID:AB_143165 | 1 to 200 |
| Antibody | anti-FLAG M2-Peroxidase | Sigma-Aldrich | A8592 RRID:AB_439702 | 1 to 3000 |
| Antibody | rabbit anti-actin | Sigma-Aldrich | A2066 RRID:AB_476693 | 1 to 2000 |
| Antibody | rabbit anti-pS6 | Cell signaling | 2215 RRID:AB_2630325 | 1 to 2000 |
| Antibody | mouse anti-GAPDH | Abcam | ab8245 RRID:AB_2107448 | 1 to 2000 |
| Antibody | HRP-anti-mouse | Dianova | 115-035-072 RRID:AB_2338507 | 1 to 6000 |
| Antibody | HRP-anti-rabbit | Dianova | 305-036-003 RRID:AB_2337936 | 1 to 6000 |
| Antibody | goat anti-rabbit IgG, AlexaFluor 488 conjugate | Life technologies | A11008 RRID:AB_143165 | 1 to 200 |
| Sequence-based reagent | primers 5'-CCC ATT CAT TGC CTT GCT GCT AGG-3' and 5'-CCT CTG GAC AGG GAA CAG | Sigma-Aldrich | custom | |
| Sequence-based reagent | siRNA AATTGACAGAG GAGTGTGATC | Qiagen | custom | |
| Sequence-based reagent | siRNA CACCGCAUCCUAGUAUCACACTT | Qiagen | custom | |

*Continued on next page*

*Continued*

| Reagent type (species) or resource | Designation | Source or reference | Identifiers | Additional information |
|---|---|---|---|---|
| Sequence-based reagent | siRNA CAGGAUUACAACUUUUAGGAATT | Qiagen | custom | |
| Sequence-based reagent | siRNA TTGAGTGAG CGCTATGACAAA | Qiagen | custom | |
| Sequence-based reagent | siRNA AAGGTGAT GAGGCTTCGCAAA | Qiagen | custom | |
| Sequence-based reagent | siRNA TAGAACGT GATGAGTCATCAT | Qiagen | custom | |
| Sequence-based reagent | non siRNA AATTCTCCG AACGTGTCACGT | Qiagen | custom | |
| Chemical compound, drug | Hoechst33342 | Sigma-Aldrich | B2261 CHEBI:51232 | 1 to 1000 |
| Chemical compound, drug | Fluoromount | Sigma-Aldrich | F4680 | |
| Chemical compound, drug | Fluoroshield Mounting Medium | Abcam | ab104135 | |
| Chemical compound, drug | PBS tabletts | gibco | 18912–014 | |
| Chemical compound, drug | Triton-X | Roth | 6683.1 CHEBI:9750 | 0.3% |
| Chemical compound, drug | Tween20 | Roth | 9127.1 | 0.1% |
| Chemical compound, drug | Triton X-100 | Sigma-Aldrich | T8787 CHEBI:9750 | 1 – 0.1% |
| Chemical compound, drug | natural donkey serum | Abcam | ab7475 RRID: AB_2337258 | 4 – 2% |
| Chemical compound, drug | natural goat serum | Abcam | ab7481 RRID:2532945 | 4 – 2% |
| Chemical compound, drug | natural sheep serum | Abcam | ab7489 RRID: AB_2335034 | 20% |
| Chemical compound, drug | xylocaine | AstraZeneca | PUN080440 | 2% |
| Chemical compound, drug | isoflurane | AbbVie | 8506 CHEBI:6015 | 1–1.55% |
| Chemical compound, drug | PBS | Life technologies | 18912–014 | 1 M |
| Chemical compound, drug | paraformaldehyde | Life technologies | 15710 CHEBI:31962 | diluted to 4% |
| Chemical compound, drug | Oregon-Green BAPTA1 AM | Molecular probes | O6807 | 1 mM |
| Chemical compound, drug | EGTA | Sigma-Aldrich | E4378 CHEBI:30740 | 0.5 mM |
| Chemical compound, drug | $MgCl_2$ | Sigma-Aldrich | M2670 CHEBI:86345 | 3 mM |
| Chemical compound, drug | K-lactobionate | Sigma-Aldrich | L2398 CHEBI:55481 | 60 mM |
| Chemical compound, drug | Taurine | Sigma-Aldrich | T0625 CHEBI:15891 | 20 mM |
| Chemical compound, drug | $KH_2PO_4$ | Sigma-Aldrich | P5655 CHEBI:63036 | 10 mM |
| Chemical compound, drug | HEPES | Sigma-Aldrich | H3375 CHEBI:42334 | 20 mM |

*Continued on next page*

*Continued*

| Reagent type (species) or resource | Designation | Source or reference | Identifiers | Additional information |
|---|---|---|---|---|
| Chemical compound, drug | Sucrose | Sigma-Aldrich | S0389 CHEBI:17992 | 110 mM |
| Chemical compound, drug | BSA | Sigma-Aldrich | A6003 | 1 g/L |
| Chemical compound, drug | Malate | Sigma-Aldrich | M1000 CHEBI:6650 | 2 mM |
| Chemical compound, drug | Pyruvate | Sigma-Aldrich | P2256 CHEBI: 50144 | 10 mM |
| Chemical compound, drug | Glutamate | Sigma-Aldrich | G1626 CHEBI:64243 | 20 mM |
| Chemical compound, drug | ADP | Sigma-Aldrich | A2754 CHEBI:16761 | 5 mM |
| Chemical compound, drug | Succinate | Sigma-Aldrich | S2378 CHEBI:63686 | 10 mM |
| Chemical compound, drug | FCCP | Sigma-Aldrich | C2920 CHEBI:75458 | 0.2 µM |
| Chemical compound, drug | Rotenone | Sigma-Aldrich | R8875 CHEBI:28201 | 0.1 µM |
| Chemical compound, drug | Antimycin A | Sigma-Aldrich | A8674 CHEBI:22584 | 2 µM |
| Chemical compound, drug | $MgSO_4$ | Sigma-Aldrich | 203726 CHEBI:32599 | |
| Chemical compound, drug | NaCl | Sigma-Aldrich | S3014 CHEBI:26710 | |
| Chemical compound, drug | $Na_2HPO_4$ | Sigma-Aldrich | S3264 CHEBI:34683 | |
| Chemical compound, drug | oligofectamine | Thermo-Fisher | 12252–011 | 0.2% |
| Chemical compound, drug | metformin | MP Biomedicals | 157805 CHEBI:6802 | 5 mg/ml |
| Chemical compound, drug | urea | Roth | 2317.3 CHEBI:16199 | 48% |
| Chemical compound, drug | Tris | Roth | 4855.2 CHEBI:9754 | 15 mM |
| Chemical compound, drug | Glycerin | Roth | 3783.1 CHEBI:17754 | 8.7% |
| Chemical compound, drug | SDS | Roth | 2326.1 CHEBI:8984 | 1% |
| Chemical compound, drug | mercaptoehanol | Roth | 4227.3 CHEBI:41218 | 1% |
| Chemical compound, drug | protease inhibitors | Roche | 04 693 116 001 | 1 tablet per 50 ml |
| Chemical compound, drug | Phosstop | Roche | 04 906 837 001 | 2 tablets per 10 ml |
| Software, algorithm | GraphPad Prism | GraphPad Prism | RRID:SCR_002798 | http://www.graphpad.com/ |
| Software, algorithm | Igor Pro 6.22 – 6.37 | Wavemetrics | RRID:SCR_000325 | http://www.wavemetrics.com/products/igorpro/igorpro.htm |
| Software, algorithm | MATLAB R2011a | Mathworks | RRID:SCR_001622 | https://www.mathworks.com |

*Continued*

| Reagent type (species) or resource | Designation | Source or reference | Identifiers | Additional information |
|---|---|---|---|---|
| Software, algorithm | Code use for Calcium transient analysis | this paper | | the code is enclosed as a source file |
| Software, algorithm | LaVision BioTec ImSpector microscopy software | LaVision BioTec | RRID:SCR_015249 | https://www.lavisionbiotec.com/ |
| Software, algorithm | Fiji | Fiji | RRID:SCR_002285 | http://fiji.sc |
| Software, algorithm | Image J | Image J - NIH | RRID:SCR_003070 | https://imagej.nih.gov/ij/ |
| Software, algorithm | EthoVision XT 8.5 | Noldus | RRID:SCR_000441 | https://www.noldus.com/EthoVision-XT/New |
| Software, algorithm | Image lab | Biorad | RRID:SCR_014210 | http://www.bio-rad.com/en-us/sku/1709690-image-lab-software |
| Software, algorithm | Oroboros DatLab | Oroboros, Innsbruck, Austria | | http://www.oroboros.at/index.php?id=datlab |

## Animals

All experimental procedures were performed in accordance with institutional animal welfare guidelines and were approved by the state government of Rhineland-Palatinate, Germany (G14-1-010 and G14-1-017). WT littermates and heterozygous $Hdh^{(CAG)150}$ mice (Hdh150, RRID:IMSR_JAX:004595) carrying an extended CAG sequence (~150) replacing the normal length CAG sequence in mouse *Htt* gene were obtained by crossing Hdh150 heterozygous with WT mice (*Lin et al., 2001*).

Male mice at 10 – 15 weeks of age were used to examine the change in neuronal network activity prior to disease onset and at 14 – 17 weeks of age to examine the effect of *in vivo* metformin treatment. Male mice at 12 – 16 weeks of age were used for behavior studies. Male mice at 13 weeks of age were used for immunohistochemistry. The mice were kept under specific-pathogen-free conditions on a 12 hr light/12 hr darkness cycle with free access to water and food. Mice were genotyped using the primers 5'-CCC ATT CAT TGC CTT GCT GCT AGG-3' and 5'-CCT CTG GAC AGG GAA CAG TGT TGG-3' (Sigma-Aldrich) producing 379- and 829-bp-long fragments for WT and mutant alleles, respectively.

## Surgery for *in vivo* two-photon Ca²⁺ imaging

Mice were prepared for *in vivo* imaging under isoflurane (1 – 1.5% in pure $O_2$, AbbVie). Anesthesia depth was assessed by monitoring pinch withdrawal and respiration rate. Body temperature was kept at 37°C with a heating pad (ATC 200, World precision instruments). Local anesthesia (2% xylocaine, AstraZeneca) was applied to the scalp. A custom-made recording chamber was glued to the skull with cyanoacrylic glue (UHU) followed by dental cement (Paladur, Heraeus). Then, a craniotomy of 1.5 × 1.5 mm was performed using stainless steel drill bits. The position of the primary visual cortex was located according to brain atlas coordinates (Bregma −3 to −4.5 mm, 2 – 3 mm lateral to the midline) (*Paxinos and Franklin, 2001*). After surgery, the mouse was subjected to the two-photon imaging setup.

## Two-photon Ca²⁺ imaging

The fluorescent Oregon-Green BAPTA1 AM (OGB-1 AM, O6807, Molecular Probes) was bulked-loaded in the visual cortex as described previously (*Stosiek et al., 2003*). Anesthesia level was continuously monitored by keeping the breathing rate at 100 – 110 breaths/min. High-speed two-

photon Ca$^{2+}$ imaging was performed in layer 2/3 (150 to 350 μm from the pial surface) with an upright LaVision BioTec TrimScope II resonant scanning microscope, equipped with a Ti:sapphire excitation laser (Chameleon Ultra II, Coherent) and a 25x (1.1 N.A., MRD77220, Nikon) or 40x (0.8 N.A., NIRAPO, Nikon) objective. The laser was tuned to 800 nm and fluorescence emission was isolated using a band-pass filter (525/50, Semrock) and detected using a GaAsP photomultiplier tube (PMT; H7422-40, Hamamatsu). The TriM Scope II scan head, equipped with a resonant scanner, imaged time-lapses (512 × 512 pixels,~440 × 440 μm field of view) at a maximum frame rate of 30.4 Hz. Time lapses were recorded for 5 – 8 min on average. Imspector software (LaVision BioTec) was used for microscope control and image acquisition.

## Determining cell number and analysis of Ca$^{2+}$ transients

First, the number of cells loaded with OGB-1 was manually counted in ImageJ (National Institutes of Health). The area containing all the cells was traced freehand and calculated by the software. Functional data were analyzed using custom-written functions in MATLAB R2011a (Mathworks, Natick, MA) and Igor Pro 6.22 – 6.37 (Wavemetrics, Inc., Lake Oswego, OR). The code is attached as *Figure 1—source data 2*. Regions of interest (ROIs) were hand-drawn by tracing the outlines of OGB1-positive neurons. Fluorescence intensities were quantified by averaging pixels inside each ROI for every image in a sequence. The fluorescence values were normalized by user-defined baseline. Specifically, dF/F was defined as the following:

$$\frac{dF}{F} = \left( \frac{mean\ fluorescence\ inside\ an\ ROI}{mean\ user-defined\ baseline} - 1 \right) * 100$$

where the baseline is defined as a mean fluorescence from a 1–3 s silent period in the same ROI. The peak of a Ca$^{2+}$ transient was defined as the first derivative to crossed zero, and the second derivative to be negative, and where the amplitude to be greater than three standard deviations (SD) above the mean. The peak location was corrected manually where necessary. Each dF/F trace, sampled at 15.2 – 30.4 Hz sampling frequency, was preprocessed by binomial Gaussian smoothing (20 – 40 iterations) followed by a high pass filter. The baseline was estimated as the median of activity-free 10 s period preceding each peak. The foot and the tail of Ca$^{2+}$ transients were determined as the first data point that fell within 0.5 SD of the baseline before and after the peak, respectively. The area under the curve was trapezoidal and measured between the foot and the tail. A distribution histogram of neurons according to their Ca$^{2+}$ transient frequency was used to segregate neurons into three functional subgroups. The definition of hyperactive neurons (>3 trans/min) was determined by the absence of neurons above this limit in WT (0.4 ± 0.3%). The criterion used for low active neurons was set to comprise ~25% of the WT neuronal population (25.8 ± 3.9%).

The distance between two neurons was calculated by Pythagorean theorem after the x,y coordinate was determined for each ROI.

The randomization of experimental data comprised two steps: first, each raster plot was reassigned to a randomly selected ROI; then, the location of the individual raster event was shuffled randomly, except no spikes were allowed to occur within 1 s of each other.

## Immunohistochemistry

Mice were anesthetized with a mixture of ketamine/xylocaine and perfused transcardially with 4% paraformaldehyde (#15710, Life technologies) in 0.1M phosphate buffer and brains were post-fixed. 50-μm-thick sections were sliced using a HM650 V vibratome (ThermoFisher) and collected in phosphate buffer saline (PBS; Life technologies). Floating sections were incubated for 1 hr with PBS containing 4% natural goat serum (NGS, ab7481, Abcam) or 4% natural donkey serum (NDS, ab7475, Abcam) and 1% Triton X-100 (Sigma-Aldrich) at room temperature (RT, 22℃). Slices were then incubated for 48 hr at 4℃ with primary antibodies against the apoptotic marker-cleaved caspase-3 (1:500; rabbit polyclonal; 9661, Cell signaling), neuronal marker NeuN (1:500; mouse monoclonal; MAB377, Millipore) or astrocytic marker GFAP (1:1500; rabbit polyclonal; Z0334, Dako). Slices were incubated for 2 hr at RT with secondary antibody Alexa 546 goat anti-rabbit (1:300, A11035, Invitrogen), Alexa 647 goat anti-mouse (1:300, A21235, Invitrogen) or Cy2 donkey anti-rabbit (711-225-152, Jackson Immuno Research). Primary and secondary antibodies were diluted in PBS containing

2% NGS or 2% NDS and 0.2% Triton X-100. After staining, brain slices were mounted with Fluoroshield Mounting Medium (ab104135, Abcam).

For Htt staining, brains were embedded in tissue tek (Sakura) and frozen on dry ice with 100% ethanol. 5 – 10 μm sagittal sections were sliced and subjected to antigen retrieval by being placed in 10 mM sodium citrate buffer at 84°C or 90 – 95°C for 15 – 20 min and rinsed with TBS-Triton-X (0.3%, Roth). Subsequently, sections were blocked with 20% sheep or horse serum in TBS-Triton-X for 1 hr at RT. Primary antibody (Htt: 1:200, rabbit monoclonal, ab109115, Abcam) was diluted in TBS-Triton-X and incubated overnight at 4°C. Secondary antibody (1:200, goat anti-rabbit AlexaFluor 488, A11008, Life technologies) in TBS-Triton-X was incubated for 2 hr at RT. Afterwards, sections were embedded in fluoromount (Sigma-Aldrich) including Hoechst33342 (1:1000, B2261, Sigma-Aldrich). Mounted slices were analyzed with a confocal laser-scanning microscope (Leica SP8).

## Sample preparation for respirometry experiments

Experimental animals were sacrificed by cervical dislocation immediately before OXPHOS analysis. Brains were micro-dissected on ice and specimens weighed on an analytical balance (Sartorius, CPA1003S; Germany). The micro-dissected brain regions were directly transferred into ice-cold mitochondrial respiration medium MiR05 (EGTA 0.5 mM, $MgCl_2$ 3 mM, K-lactobionate 60 mM, taurine 20 mM, $KH_2PO_4$ 10 mM, HEPES 20 mM, sucrose 110 mM, BSA 1 g/L, adjusted to pH 7.1) (*Kuznetsov et al., 2000*). The tissue was then homogenized in a pre-cooled 1.5 ml tube with a motorized pestle in MiR05 medium with 10 strokes. Resulting homogenates containing 10 mg tissue wet weight were suspended in 100 μl of ice-cold MiR05 and later 20 μl (2 mg) from the 100 μl tissue suspension was added to each chamber of the Oxygraph-2k, Oroboros Instrument containing 2 ml of MiR05 for OXPHOS analysis (*Holmström et al., 2012*). All chemicals were purchased from Sigma-Aldrich, Germany. The optimized motorized pestle preparation of brain tissue yields a high degree of permeabilization as evident by the minimal effect of digitonin titrations on OXPHOS capacity. Therefore, digitonin is not necessary for this protocol.

## High-resolution respirometry in tissue

Tissue homogenates were transferred into calibrated Oxygraph-2k (O2k, Oroboros Instruments, Innsbruck, Austria) 2 ml chambers. Oxygen polarography was performed at 37 ± 0.001°C (electronic Peltier regulation) in O2k chambers and oxygen concentration (μM) as well as oxygen flux per tissue mass (pmol $O_2 \cdot s^{-1} \cdot mg^{-1}$) were recorded real-time using DatLab software (Oroboros Instruments Innsbruck, Austria). A multisubstrate protocol was used to sequentially explore the various components of mitochondrial respiratory capacity. The homogenate was suspended in MiR05, added to the Oxygraph-2k glass chambers and the $O_2$ flux was allowed to stabilize. A substrate-uncoupler-inhibitor titration (SUIT) protocol was applied to assess qualitative and quantitative mitochondrial changes in Hdh150 transgenic mice and unaffected controls. After stabilization, LEAK respiration was evaluated by adding the complex I (CI) substrates malate (2 mM), pyruvate (10 mM) and glutamate (20 mM). The maximum oxidative phosphorylation (OXPHOS) capacity with CI substrates was attained by the addition of ADP+$Mg^{2+}$ (5 mM) ($CI_{OXPHOS}$). For evaluation of maximum OXPHOS capacity of the convergent input from CI and complex II (CII) at saturating ADP-concentration, the CII substrate succinate (10 mM) was added (CI +$CII_{OXPHOS}$). We then uncoupled respiration to examine the maximal capacity of the electron transport system (ETS or CI +$II_{ETS}$) using the protonophore, carbonylcyanide 4 (trifluoromethoxy) phenylhydrazone (FCCP; successive titrations of 0.2 μM until maximal respiration rates were reached). We then examined consumption in the uncoupled state solely due to the activity of complex II by inhibiting complex I with the addition of rotenone (0.1 μM; ETS CII or $CII_{ETS}$). Finally, electron transport through complex II was inhibited by adding antimycin A (2 μM) to obtain the level of residual oxygen consumption (ROX) due to oxidating side reactions outside of mitochondrial respiration. The $O_2$ flux obtained in each step of the protocol was normalized by the wet weight of the tissue sample used for the analysis and in addition ROX was subtracted from the fluxes in each run to correct for non-mitochondrial respiration (*Hollis et al., 2015*). All respiration experiments comprise 2 – 3 counterbalanced blocks across days. All substrates and inhibitors used were obtained from Sigma.

## Visual discrimination task

WT and presymptomatic VFDO Hdh150 mice (13 – 15 weeks of age) were isolated and food deprived for 24 hr. Subsequently, they were placed into an operant chamber with a touchscreen including two windows and a food dispenser on the opposite wall (Med Associates Inc; St. Albans). In order to keep animals motivated to perform the task, their daily food intake was adjusted to maintain body weight at 75 – 80% of their initial body weight during the course of the experiment. The experiment consisted of three phases:

1) mice were trained to collect a food pellet from the dispenser twice on day 1, 2) mice were trained to nose poke the touchscreen to obtain food pellet reward. They needed to collect and consume the pellet to proceed to the next trial. One daily session was either 30 min or 70 trials. The touch training was over when mice reached 70 trials on three consecutive days. 3) For the visual discrimination task, the screen presented two stimuli (pair 1: black vs. white or pair 2: black vs. grey), one correct, one false, randomly presented left or right. The mice were trained to nose poke the correct stimulus, whereupon a pellet was released. Again, they needed to collect and consume the pellet to proceed to the next trial. One daily session was either 30 min or 100 trials. The task was successful when the mice reached 70% correct trials on three consecutive days.

## Open field test

Mice were not habituated to the set-up. Each mouse was removed from its home cage and put into a holding box next to the testing box. Subsequently, the mouse was put into the testing box facing the rear wall. The mice had time to explore the area for 5 min. Time in the center, which was determined as 10 cm away from each wall of the box, was measured automatically by EthoVision XT 8.5, when the center-point of the mouse moved into it.

## Cell lines and filter retardation assay

For all cell lines used in this study the identity has been authenticated and mycoplasma contamination has been tested and excluded.

HEKT cell lines (ATCC, RRID:CVCL_1926) stably expressing FLAG-tagged HTT-exon 1 with either 51 or 83 CAG repeats under the control of a Tet-off promotor as well as the filter retardation assay were described previously (*Scherzinger et al., 1997*).

For the filter retardation assay cells were either transfected with a pool of MID1 specific siRNA oligonucleotides (AATTGACAGAGGAGTGTGATC, CACCGCAUCCUAGUAUCACACTT, CAGGA UUACAACUUUUAGGAATT, TTGAGTGAGCGCTATGACAAA, AAGGTGATGAGGCTTCGCAAA, TAGAACGTGATGAGTCATCAT) or non-silencing control oligonucleotides (AATTCTCCGAACGTG TCACGT) using Oligofectamine (Thermo Fisher Scientific) or treated with metformin at a final concentration of 1 mM and 2.5 mM for 24 hr. Cell lysates were soaked through a filter membrane and aggregates were detected using monoclonal anti-FLAG M2-Peroxidase (HRP) antibodies (Sigma-Aldrich). Signals were quantified using the Fiji Software.

## FRAP (fluorescence recovery after photobleaching)-based assay

Neuro-2A (a mouse neuroblastoma cell line, ATCC, RRID:CVCL_0470) cells or murine primary cortical neurons (prepared from NMRI mice E14.5 as described previously [*Kickstein et al., 2010*] were transfected with constructs expressing Htt exon1 with 49CAG repeats fused to GFP (vector pEGFP-C1-HTTex1; an N-terminal GFP tag) the day before analysis. Cells were analyzed in a previously established FRAP-based assay to monitor translation in living cells (*Krauss et al., 2013*) using a Zeiss LSM700 confocal microscope. In brief, in contrast to standard FRAP, the GFP-signal of the entire cell was bleached using a 488 nm argon laser and fluorescence recovery was imaged over a time frame of 4 hr. The fluorescence signal was quantified as the sum of the pixel over the cell area, and the resulting total cell fluorescence was normalized to the post-bleach signal, which was set to 100%. Fluorescence recovery curves represent mean ± SEM of at least 35 cells.

## Caenorhabditis elegans

The following *C. elegans* strains was used: strain AM141, genotype rmIs133 [unc-54p::Q40::YFP] (RRID:WB-STRAIN:AM141). AM141 worms express YFP that is linked to a polyglutamine stretch of 40 glutamines (Q40) in the muscle cells of the body wall. In the early lifetime of the worms YFP-Q40

is soluble and it aggregates gradually over time. This strain is used as a model for polyglutamine diseases like Huntington´s disease, since the 40Q represents a pathological range of the polyglutamine stretch.

For the treatment with metformin, NGM plates were seeded with OP50 bacteria and dried overnight. For heat inactivation OP50 bacteria were incubated at 70°C for 30 min. Metformin in a concentration of 5 mM, 10 mM or 500 mM was added the next day and dried again before usage. Worms were then put onto the plates and aggregates and liquid trashing were quantified after 5 days.

Nematodes were synchronized by hypochloride treatment. At day 5 of adulthood the worms were analyzed. Aggregates were counted under a fluorescence stereo microscope after anesthetizing the worms with 25 mM Levamisol on a coverslip. For each experiment 15 – 20 animals were counted. In addition, liquid thrashing was analyzed in 10 – 15 animals per experiment. Therefore, single worms were transferred into one drop of M9 buffer (3 g $KH_2PO_4$, 6 g $Na_2HPO_4$, 5 g NaCl, 1 ml 1 M $MgSO_4$, $H_2O$ to 1 liter) and the rhythmic bending of the worm around its body axis was counted for 30 s. Each experiment was conducted at least three times.

## Western blots

For western blotting mice were sacrificed and brains were grinded and shaked in magic mix (48% urea, 15 mM TRIS-HCl (pH7.5), 8.7% glycerin, 1% SDS, 1% mercaptoethanol, complete protease inhibitors (Roche), Phosstop (Roche) at 4°C with add-on homogenization (QIAshredder). Afterwards, samples were boiled at 95°C and 30 μg protein lysate (40 μg for Htt blot) was loaded on a 10% SDS PAGE gel, resolved (overnight at 100 V for separation of mHtt from wtHtt) and blotted onto a PVDF membrane (BioRad) using TransBlot Turbo (BioRad). Membranes were then blocked with 1% BSA (pS6) or 5% milk (Actin, Gapdh, Htt) in PBS-Tween20 (Roth) and incubated with primary antibody (Actin: 1:2000, A2066, Sigma Aldrich; pS6: 1:2000, 2215, cell signaling; Gapdh: 1:2000, ab8245, abcam; Htt: 1:850, ab109115, abcam) in blocking buffer overnight. Membranes were then washed three times with PBS-Tween20 and subsequently incubated with secondary antibody (1:6000, for Htt 1:4000, Donkey anti-rabbit or anti-mouse, Jackson Immuno Research) for 1 hr in blocking buffer. Subsequently, membranes were again washed three times with PBS-Tween20. Chemiluminescent detection was done by using Western Lightning Plus-ECL (PerkinElmer). Visualisation was performed on a ChemiDoc MP Imaging System (Biorad). Quantification of resulting bands was performed using Image Lab (version 5.2.1).

## Metformin treatment in vivo

Both the Hdh150 and WT mice received chronic metformin (MP Biomedicals, LLC; France) administration (5 mg/ml in the drinking water) freshly prepared every day for 3 weeks starting from an age of 9 – 10 weeks.

## Statistics

Statistical significance was tested in GraphPad Prism (GraphPad Software Inc., La Jolla, CA). *Table 1* contains all details concerning statistical tests used (name of the test, p-values, F values and degree of freedom). $*p < 0.05$, $**p < 0.01$, $***p < 0.001$ and $****p < 0.0001$. For all data, we first tested for normal distribution using the one-sample Kolmogorov-Smirnov test. In case that the null hypothesis of a normal distribution could not be rejected (for $p > 0.05$), we employed a parametric test, if $H_0$ could be rejected (for $p < 0.05$) we used non-parametric tests: Mann-Whitney U test for non-parametric data and t test for parametric data. Pearson's correlation coefficient was used on raster plots that were temporally binned (328 ms per bin) to compare activity patterns between pairs of neurons. Box-and-whisker plots indicate the median (line) of average values from multiple time-lapses, the 25-75th percentiles (box) and the 10-90th percentiles (whiskers). Graphs show mean ±s.e.m (standard error of the mean).

### Data availability

Values used in figures are available on Dryad Digital repository (doi:10.5061/dryad.g3b5272) and the code used for the analysis of calcium imaging is in *Figure 1—source data 2*.

## Acknowledgement

The authors thank all members of the Methner, Schweiger and Stroh labs for advice and helpful discussions. We thank Zeke Barger and Andrea Kronfeld for help in the analysis of two-photon $Ca^{2+}$ imaging data. We also thank Ulrich Schmidt for help in behavior experiments and Cheryl Ernest, Ina Vorberg, Dan Ehninger, Simon Rumpel and Oliver Tuescher for proofreading the manuscript. We acknowledge the Caenorhabditis Genetics Center for the worm strain.

## Additional information

### Funding

| Funder | Author |
| --- | --- |
| European Huntington's Disease Network | Axel Methner<br>Albrecht Stroh |
| Focus Program Translational Neuroscience | Isabelle Arnoux<br>Michael Willam<br>Axel Methner<br>Susann Schweiger<br>Albrecht Stroh |
| BMBF Eurostars | Isabelle Arnoux<br>Albrecht Stroh |
| Tenovus | Jeremy J Lambert |

The funders had no role in study design, data collection and interpretation, or the decision to submit the work for publication.

### Author contributions

Isabelle Arnoux, Michael Willam, Data curation, Formal analysis, Investigation, Visualization, Writing—original draft, Writing—review and editing; Nadine Griesche, Data curation, Formal analysis, Investigation, Visualization, Writing—original draft; Jennifer Krummeich, Investigation; Hirofumi Watari, Data curation, Software, Formal analysis, Methodology, Writing—original draft; Nina Offermann, Stephanie Weber, Formal analysis, Investigation; Partha Narayan Dey, Data curation, Formal analysis, Investigation, Visualization; Changwei Chen, Olivia Monteiro, Data curation, Formal analysis, Methodology; Sven Buettner, Katharina Meyer, Daniele Bano, Data curation, Formal analysis, Investigation; Konstantin Radyushkin, Resources, Formal analysis, Supervision, Methodology; Rosamund Langston, Formal analysis, Supervision, Methodology; Jeremy J Lambert, Supervision, Methodology; Erich Wanker, Data curation, Supervision; Axel Methner, Conceptualization, Data curation, Formal analysis, Supervision, Funding acquisition, Writing—original draft, Writing—review and editing; Sybille Krauss, Conceptualization, Data curation, Formal analysis, Supervision, Investigation, Writing—original draft, Writing—review and editing; Susann Schweiger, Albrecht Stroh, Conceptualization, Data curation, Supervision, Funding acquisition, Writing—original draft, Writing—review and editing

### Author ORCIDs

Isabelle Arnoux (iD) http://orcid.org/0000-0003-4530-9944
Erich Wanker (iD) http://orcid.org/0000-0001-8072-1630
Axel Methner (iD) https://orcid.org/0000-0002-8774-0057
Albrecht Stroh (iD) http://orcid.org/0000-0001-9410-4086

### Ethics

Animal experimentation: All experimental procedures were performed in accordance with institutional animal welfare guidelines and were approved by the state government of Rhineland-Palatinate, Germany (G14-1-010 and G14-1-017).

### Decision letter and Author response

Decision letter https://doi.org/10.7554/eLife.38744.031

Author response https://doi.org/10.7554/eLife.38744.032

## Additional files

**Supplementary files**
• Transparent reporting form
DOI: https://doi.org/10.7554/eLife.38744.027

**Data availability**

Values used in figures are available on Dryad Digital repository (doi:10.5061/dryad.g3b5272). The code used for the analysis of calcium imaging is attached as a source file. The numerical values for each figure are enclosed as source data files.

The following dataset was generated:

| Author(s) | Year | Dataset title | Dataset URL | Database, license, and accessibility information |
|---|---|---|---|---|
| Arnoux I, Willam M, Griesche N, Krummeich J, Watari H, Offermann N, Weber S, Narayan Dey P, Chen C, Monteiro O, Buettner S, Meyer K, Bano D, Radyushkin K, Langston R, Lambert JJ, Wanker E, Methner A, Krauss S, Schweiger S, Stroh A | 2018 | Data from: Metformin reverses early cortical network dysfunction and behavior changes in Huntington's disease | https://dx.doi.org/10.5061/dryad.g3b5272 | Available at Dryad Digital Repository under a CC0 Public Domain Dedication |

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
