## [Decision Letter]

Thank you for submitting your article "Metformin reverses early cortical network dysfunction and behavior changes in Huntington's disease" for consideration by *eLife*. Your article has been reviewed by three peer reviewers, one of whom is a member of our Board of Reviewing Editors, and the evaluation has been overseen by a Senior Editor. The following individual involved in review of your submission has agreed to reveal their identity: Joan S Steffan (Reviewer #2).

The reviewers discussed the reviews with one another and the Reviewing Editor drafted this decision to help you prepare a revised submission.

Summary:

All reviewers found that this work nicely describes development of an assay to measure alterations in neuronal microcircuits in HdhQ150 knock-in mice at the premanifest stage. The authors show that at an early age of 10-15 weeks, the cortical microcircuit of heterozygous HdhQ150 knock-in HD mice shifts towards a more excitable state with increased synchronicity. As the authors had previously shown that the MID1/PP2A/mTOR protein complex regulates translation efficiency of HTT mRNA in a CAG repeat-dependent manner, and that metformin interferes with the MID1 complex, they show that metformin reduces mutant huntingtin (HTT) translation. The authors went on to determine that metformin reverses signs of HD pathology in vivo. First, they used a *C. elegans* model expressing YFP-tagged Q40 polypeptide in body wall muscle cells and determined that metformin reduced the number of intracellular inclusion bodies and rescued motility impairment, similar to knockdown of Arc-1, the *C. elegans* homolog of MID1. Second, they found that metformin treatment in the drinking water of VFDO HdhQ150 mice completely restored the alterations in cortical microcircuit activity back to wt levels.

Recent work by the laboratory of Gillian Bates (PNAS 110:266, 2013 and Sci Rep 7, 1307, 2017) has demonstrated that there is incomplete transcription of the mutant HTT gene in knock-in mice and in HD patients, that results in expression of mutant HTT exon 1 protein (mHTTex1p), known to cause Huntington's disease-like pathogenesis in transgenic mice and flies. mHTTex1p has been shown to be cleared by autophagy, a lysosomal process of protein degradation known to decline with age. Full-length HTT itself is an autophagic scaffold, and its levels decline with age in both wt and mutant HD knock-in mice. The ideal therapeutic for HD may be to reduce translation of mHTTex1p in patients in parallel with early presymptomatic activation of autophagy to clear mHTTex1p while lysosomal function is intact, without reducing levels of wt full-length HTT which may be protective. The authors of this manuscript demonstrate that mHTTex1p translation is reduced with metformin treatment, and that in HdhQ150 heterozygotes, only mutant full-length HTT is reduced, but that wt full-length HTT levels are not impacted, although the promising western shown in 4f needs to be redone in triplicate with loading controls added. The authors state that the effect of metformin at the dose they use does not activate AMPK, known to activate autophagy; activation of AMPK-induced autophagy has been found to improve HD pathology in cell culture (Walter 2016). Reduced mTor activity, shown here with metformin treatment of HdhQ150s, is a well-known activator of autophagy. Metformin has also previously been shown by others to activate autophagy, to slow HD pathology in male R6/2 HD transgenic mice, and to improve cognitive function in diabetic HD patients. As the authors point out, metformin is inexpensive and well known in long-term clinical usage with only benign side effects. Metformin might be a wonderful treatment for HD to be tested in gene-positive premanifest patients, as it may reduce levels of toxic mHTTex1p through reduced translation as well autophagy activation, while not affecting wt full-length HTT levels or function. This work is very promising but needs revision on several points.

Essential revisions:

1) Female mice were excluded from this study without rationale. In the Discussion, the authors state that Ma et al. reported that only male R6/2 mice benefitted from metformin treatment. These sex-specific differences should be investigated more thoroughly. If this treatment is expected to only be beneficial for males, this needs to be clearly addressed in the text.

2) Experiments on the behavioral deficits in the mice are not well done. If these mice have a memory consolidation deficit, they should not learn the visual discrimination task. The authors are putting forth that these mice have perfect consolidation in visual training and none in object recognition, a late onset deficit. It is far more likely that these mice have anxiety, a known early onset phenotype in patients and animals, and their neophobia confounds object learning assessment. The test was also performed using objects that the mice could climb on. This is a confound and data from these types of objects is skewed. All in all, this data should be removed and cognitive effects of metformin not addressed in this paper.

3) The finding that metformin induces mHTT lowering is not convincing. A filter retardation assay demonstrates reduced aggregation, but decreased aggregation does not equal decreased mHTT. The Western image in Figure 4E is unquantifiable. If this is representative, the quantification is not valid. Additionally, instead of a loading control, mHTT is normalized to wt HTT, which pre-supposes perfect specificity for mHTT. Finally, it is suggested that authors eliminate 4E and focus on improving data in Figure 4F by including multiple replicates and a loading control at the later time point. Finally, if metformin does reduce mHTT, this study reports previously unexplored deficits that, like many others, are ameliorated by HTT lowering rather than a basis for a paradigm shift, in which rebalancing network function represents a novel therapeutic strategy for HD. This latter point needs to be presented in their Discussion.

4) The authors need to expand their review of the background literature in HD to include more on HTT loss in vivo. Molero et al., showing that expression of full-length mHTT only during pre-natal development is sufficient to induce HD-like deficits in adult mice, is cited as rationale for the importance of very early intervention. However, Ai Yamamoto's seminal work demonstrating that adult inactivation of mHTTex1p expression in symptomatic mice leads to a reversal of behavior deficits and neuropathology (Yamamoto et al., 2000), and Gill Bates's recent data demonstrating that mHTTex1p is expressed due to incomplete transcription of the mutant HTT gene in patients, suggest that reduced translation of mHTTex1p in patients may be helpful to slow disease onset. Furthermore, based on cited studies, the authors conclude that postnatal HTT loss does not cause adverse effects, but do not include papers demonstrating that perinatal or adult inactivation of HTT results in a progressive neurodegenerative phenotype and that older animals are more susceptible to this effect (Dragatsis et al., 2000, Dietrich et al., 2017).

---

## [Author Response]

Essential revisions:1) Female mice were excluded from this study without rationale. In the Discussion, the authors state that Ma et al. reported that only male R6/2 mice benefitted from metformin treatment. These sex-specific differences should be investigated more thoroughly. If this treatment is expected to only be beneficial for males, this needs to be clearly addressed in the text.

While we conducted an a priori power analysis to determine the sample size based on previous studies on network dysfunction in early-stage Alzheimer’s disease, this is the very first study on cortical microcircuit activity in early stage Huntington’s disease models in vivo. Consequently, the effect size was unknown and could only be approximated. To increase the chances of detecting the hypothesized effect, we chose male mice, thereby minimizing the influence of hormonal fluctuations on network activity. This approach is in line with a recent study in the field of Alzheimer’s disease (Iaccarino et al., 2016), using males only. However, a related paper from another group used female mice, and found a similar network dysregulation in Alzheimer’s disease (Keskin et al., 2017). And the first study on this topic did not even specify the gender used (Busche et al., 2008). Consequently, while we very much agree with the reviewers on the importance of investigating gender specificity, we have no reason to assume, that the phenomenon of network dysregulation itself is limited to male mice only.

In terms of the gender-specific effects of Metformin treatment, we are aware of the Ma et al. study showing benefits of metformin in male mice only. However, Ma and colleagues have used the R6/2 mouse for their analysis and have started the treatment at 5 weeks of age, which is already late in disease progression (motor phase). The R6/2 mouse is an aggressive model and starts developing motor aberrations already in week 4 (the Hdh mouse model we employed shows first motor aberrations in week 100). We hypothesized that very early treatment in the very far from disease onset phase rather than in later phases is necessary to gain stable effects on the disease phenotype. This is supported by gene suppression studies in animal models of SCA1 (spinocerebellar ataxia I) and Huntington’s disease (summarized in Rubinsztein and Orr, Bioessays 2016) showing that only early intervention leads to stable effects in the animals. The effect sizes are highly significant and include all phenotypic features investigated. Consequently, while we fully acknowledge that there may be additional physiological variability, currently, also in terms of treatment, we do not expect that the treatment will be effective in males only. We very much agree that there needs to be a follow-up study, investigating best dosages and time of metformin treatment, which critically needs to include the investigation of gender specificity. We modified the manuscript accordingly, to make it clear, why our study was conducted in males and included a paragraph on gender specificity in the Materials and methods as well as in the Discussion section.

“We focused our analysis on male mice, thereby minimizing the influence of hormonal fluctuations on network activity. This approach is in line with a recent study in the field of Alzheimer disease (Iaccarino et al., 2016), using males only.”

“In a study on R6/2 animals, a very aggressive model for Huntington’s disease, Ma and colleagues had found a significant effect on survival rates and hind clasping in male animals only when given metformin in the drinking water starting from week 5 (Ma et al., 2007). […] While we focused on male mice in this study, to reduce physiological variability due to hormone fluctuations, at the VFDO stage brain barrier changes are not expected to influence bioavailability of metformin.”

2) Experiments on the behavioral deficits in the mice are not well done. If these mice have a memory consolidation deficit, they should not learn the visual discrimination task. The authors are putting forth that these mice have perfect consolidation in visual training and none in object recognition, a late onset deficit. It is far more likely that these mice have anxiety, a known early onset phenotype in patients and animals, and their neophobia confounds object learning assessment. The test was also performed using objects that the mice could climb on. This is a confound and data from these types of objects is skewed. All in all, this data should be removed and cognitive effects of metformin not addressed in this paper.

We have followed the advice of the reviewers and have omitted the data from the manuscript.

3) The finding that metformin induces mHTT lowering is not convincing. A filter retardation assay demonstrates reduced aggregation, but decreased aggregation does not equal decreased mHTT. The Western image in Figure 4E is unquantifiable. If this is representative, the quantification is not valid. Additionally, instead of a loading control, mHTT is normalized to wt HTT, which pre-supposes perfect specificity for mHTT. Finally, it is suggested that authors eliminate 4E and focus on improving data in Figure 4F by including multiple replicates and a loading control at the later time point. Finally, if metformin does reduce mHTT, this study reports previously unexplored deficits that, like many others, are ameliorated by HTT lowering rather than a basis for a paradigm shift, in which rebalancing network function represents a novel therapeutic strategy for HD. This latter point needs to be presented in their Discussion.

We thank the reviewers for their valuable comment.

We exchanged the western blot shown in Figure 4G, and in Figure 4—figure supplement 1. As suggested by the reviewers, we included Gapdh as loading control. Three analyses of protein level were performed: (i) mHtt/wtHtt (Figure 4H), (ii) mHtt/Gapdh (Figure 4I) and (iii) wtHTT/Gapdh (Figure 4J). Data in Figure 4H-J are from three biological replicates.

We found that the level of mHtt significantly decreases in Hdh150 met mice when normalized to Gapdh while the level of wtHtt is unchanged by metformin treatment. This indicates that metformin reduces mHtt protein load.

In addition, our findings of an early network activity shift suggests that, rather than exploring single pathomechanistical pathways for target-identification, we might also consider re-balancing the network as a promising route for future therapy development.

We have substantially changed the wording as follows: “Our data indicate that metformin treatment reverses all cortical network dysregulations in vivo in the premanifest VFDO Hdh150 mice including functional sub-group distribution, frequency and synchronicity. […] We therefore propose a shift in experimental treatment strategies: rather than exploring single pathways for target-identification we might also consider re-balancing network function in the VFDO stages of the disease.”

4) The authors need to expand their review of the background literature in HD to include more on HTT loss in vivo. Molero et al., showing that expression of full-length mHTT only during pre-natal development is sufficient to induce HD-like deficits in adult mice, is cited as rationale for the importance of very early intervention. However, Ai Yamamoto's seminal work demonstrating that adult inactivation of mHTTex1p expression in symptomatic mice leads to a reversal of behavior deficits and neuropathology (Yamamoto et al., 2000), and Gill Bates's recent data demonstrating that mHTTex1p is expressed due to incomplete transcription of the mutant HTT gene in patients, suggest that reduced translation of mHTTex1p in patients may be helpful to slow disease onset. Furthermore, based on cited studies, the authors conclude that postnatal HTT loss does not cause adverse effects, but do not include papers demonstrating that perinatal or adult inactivation of HTT results in a progressive neurodegenerative phenotype and that older animals are more susceptible to this effect (Dragatsis et al., 2000, Dietrich et al., 2017).

This paragraph includes numerous very good suggestions that helped us to substantially strengthen the manuscript. We have included all of them and thank the reviewers for their suggestions:

We have included Yamamoto’s work in the Discussion:

“Phenotype reversal could be demonstrated in a tetracyclin dependent conditional mouse model for Huntington’s disease. Both, neuropathological findings and behavior aberrations were found to disappear when mHtt protein production was stopped through a tet-off regulation mechanism in the adult animal (Yamamoto, Lucas, and Hen 2000).”

We have included Gill Bates’ work in the Discussion:

“Furthermore, a short N-terminal fragment of the mHtt protein, mHttexp1p, that is produced by incomplete exon 1 splicing and a short poly-adenylated mRNA in several animal models as well as in Huntington’s disease patients rather than full-length mHtt protein was found to be particularly pathogenic. […] This short mRNA is difficult to target by oligonucleotide strategies (Neueder et al. 2017, Sathasivam et al. 2013)”

We have strengthened the importance of the observation that metformin targets specifically the mutant Htt by discussing the work of Dragatsis et al. and Dietrich et al.: Discussion section:

“The effect of Htt loss on brain function is still under debate. SiRNA studies suggest that postnatal reduction of endogenous Htt protein is well tolerated (summarized in (Keiser, Kordasiewicz, and McBride 2016). […] Likewise, depletion of Htt protein in the adult brain leads to progressive behavior deficits (Dietrich et al., 2017)”